# Lesser Antilles Seismotectonic Zoning Model for Seismic Hazard Assessment

Océane Foix[1,2], Stéphane Mazzotti[2], Hervé Jomard[3], Didier Bertil[4], Lesser Antilles Working Group[+]

[1]Univ. Grenoble Alpes, Univ. Savoie Mont Blanc, CNRS, IRD, UGE, ISTerre, Grenoble, France.
5  [2]Géosciences Montpellier, Université de Montpellier, CNRS, Université des Antilles, France.
[3]Institut de Radioprotection et Sûreté Nucléaire, PSE-ENV, SCAN, BERSSIN, Fontenay-aux-Roses, France.
[4]Bureau de Recherche Géologique et Minière, Orléans, France.
[+]Full list of authors is detailed at the end of the paper.

*Correspondence to*: Océane Foix (oceane.foix@univ-grenoble-alpes.fr / oceane.foix@umontpellier.fr)

10  **Abstract.** Subduction zones pose a considerable challenge within the realm of seismotectonics, owing to their faults and structures interactions. The Lesser Antilles arc is a good example of how these complexities impact seismic hazard studies with a strong along-strike variations in tectonic, seismic, and volcanic activities. While they have generated significant damages, the 1839 and 1843 event characteristics (locations, depths, mechanisms, magnitudes) remain a subject of debate along with their potential implications in the megathrust seismicity, in particular in the frame of low interseismic coupling. 15  This study is grounded in the compilation of instrumental and historical seismicity, and fault catalogs, completed by analyses of focal mechanisms and rupture types as well as geodetic velocities and strain rates. The resulting seismotectonic zoning model of the Lesser Antilles encompass the upper plate, subducting oceanic plate, subduction interface, mantle wedge, and volcanoes. We propose a better depth resolution, resulting from recent studies on slab top and upper plate bottom geometries, a specific area source for the Marie-Galante graben, new propositions for mantle wedge and volcanic zoning, 20  and fully revised area sources for the subduction interface. Our study highlights specific needs for a better seismic hazard assessment in this region.

# 1 Introduction

Subduction zones are among the most complex tectonic systems due to their intricate geodynamics and the multiple interacting faults and structures affecting the subducting plate, the megathrust, and the upper plate and volcanic arc (**S1**). They are also the locus of the great M~8+ megathrust earthquakes and their disastrous effects. Because of their extraordinary nature, these megathrust earthquakes are the main focus of subduction seismicity and seismic hazard studies (e.g., Graham et al., 2021; Wirth et al., 2022). Yet, seismic hazard in subduction zones results from several different sources, whose contributions depend strongly on the considered locations and hazard spectral period (e.g., Frankel et al., 2015), thus requiring specific hazard modeling techniques (Pagani et al., 2020a). The Lesser Antilles are a good example of the difficulties associated with subduction zone seismicity and seismic hazard studies, with additional issues due to the slow deformation rate of the arc, the limited land instrumentation coverage, and the multi-country situation. The characteristics of large damaging earthquakes, such as the 1839 (M=7.5-8) and 1843 (M=8-8.5), remain debated in terms of locations, depths, mechanism, or magnitudes (Bernard and Lambert, 1988; Feuillet et al., 2011a; Hough, 2013; van Rijsingen et al., 2021). Seismic hazard models are few (Bozzoni et al., 2011), generally at the scale of the whole Caribbean region with a small focus on the Lesser Antilles (Pagani et al., 2020b; Zimmerman et al., 2022). Yet, recent studies provide important new constraints on the Lesser Antilles seismotectonics and highlight key issues for seismic hazard in subduction zones, such as the very low present-day interseismic coupling of the megathrust (van Rijsingen et al., 2021), in disagreement with Philibosian et al. (2022) hypothesis, and its implications for seismic hazard.

Estimation of the probabilistic seismic hazard assessment (PSHA) is based on (*) identifying earthquake sources, (*) characterizing their magnitude-frequency distributions, (*) the distribution of source-to-site distances and (*) predicting ground motion intensity (Baker et al., 2021). In this study, we focus on the first step by determining earthquake area and fault sources of the Lesser Antilles arc (hereafter "seismotectonic zoning model"). Previous PSHA of the French islands was conducted in 2001 (Martin and Combes, 2001). This seismotectonic zoning model and resulting PSHA calculation were, as this study, carried out in response to a request from government authorities. They both serve as a basis for the national seismic zoning revision process. The Martin and Combes (2001) crustal zoning was primarily based on structural data (gravimetric, magnetic, geologic, seismic and topo-bathymetric data from 0 to 30 km depth). The subducting plate zoning was based on plate interface dip variations induced by the presence of ridges and fractures. No specific zoning for the plate interface was proposed. Previous seismotectonic zoning models of the Lesser Antilles did not consider mantle wedge seismicity and did not integrate specific zoning for the volcanic seismicity (Martin and Combes, 2001; Pagani et al., 2020b; Zimmerman et al., 2022). Due to its physical properties, the mantle wedge is generally not considered as a site for seismic nucleation, with only weak and diffuse seismicity (Hasegawa et al. 2009). However, recent works indicate more sustained activity (New Zealand, Davey and Ristau (2011); Japan, Uchida et al. (2010); Alps, Malusa et al. (2016); Lesser Antilles, Laigle et al., (2013)). The Mw=4.5 in New-Zealand (Davey and Ristau, 2011) and a possible 1974 M=6.9-7.5 event in the

Lesser Antilles (McCann et al., 1982) raise the question of the importance of considering this seismicity for seismic hazard assessment. Allocating this seismicity to the other seismogenic sources (i.e. upper plate crust, lower plate, or the subduction interface) results in biased hypocenter distances hazard calculation. Moreover, working on PSHA in volcanic regions is a challenge regarding earthquake characteristics: low magnitude and high seismic wave attenuation. In the Lesser Antilles, the

Nevis crisis of 1950-51 caused damage to buildings, with a maximum magnitude of Mw=4.3 (ISC catalog) and intensity VIII (Willmore, 1952). It is therefore important to be able to propose a way to consider it.

In this study, we present a new seismotectonic zoning model built for Lesser Antilles, with seismogenic source characteristics provided for future seismic hazard assessment and subject to a detailed technical report (Foix et al., 2023a)

requested from the government. The seismotectonic zoning model comprise the Lesser Antilles upper plate, subducting oceanic plate, subduction interface, mantle wedge, and volcanoes, based on a compilation and reanalysis of seismicity and fault catalogs, earthquake focal mechanisms, and geodetic data. Here, we present the main scientific points leading to the zoning model, as well as a focus on the Marie-Galante graben where we estimate and compare the extension rates from seismic and geodetic data. The zoning model provides earthquake rates and maximum magnitudes, including uncertainties

and multiple options associated with the current state of knowledge. We also provide recommendations on improvements necessary for future seismotectonic and seismic hazard studies.

## 2 Lesser Antilles geodynamics

The Lesser Antilles is the result of the subduction of the North and South American Plates beneath the Caribbean Plate at a present-day convergence rate of ~20 mm/yr (DeMets et al., 2010) (**Fig. 1A, 1B**). Due to the trench convex shape, the

convergence direction is almost arc-perpendicular south of Guadeloupe and becomes more oblique to the north. The Lesser Antilles subduction is bounded its northern end by the E-W *en-échelon* strike-slip Anegada Passage system (Laurencin et al., 2017) and at its southern end by the strike-slip El-Pilar fault (Mann et al., 1991). Its western limit is marked by the Grenada Basin separating the active volcanic arc from the Aves Ridge.

The subducting plate seafloor is marked by numerous fracture zones and ridges affecting the accretionary wedge, the upper plate tectonics, and the megathrust seismogenic behavior (Pichot et al., 2012; Ezenwaka et al., 2022 and references therein). The Barracuda and Tiburon Ridges (**Fig. 1B**) mark the limit between the South and North American Plates, with convergence at the Barracuda ridge (Patriat et al., 2011), and a N-S difference in the oceanic crust thickness (Kopp et al., 2011; Laurencin et al., 2018) and fracturing (**Fig. 1B**). The Wadati-Benioff seismicity is also impacted by the presence of the

subducted fractures and ridges, as well as a possible slab tear (Harris and al., 2018).

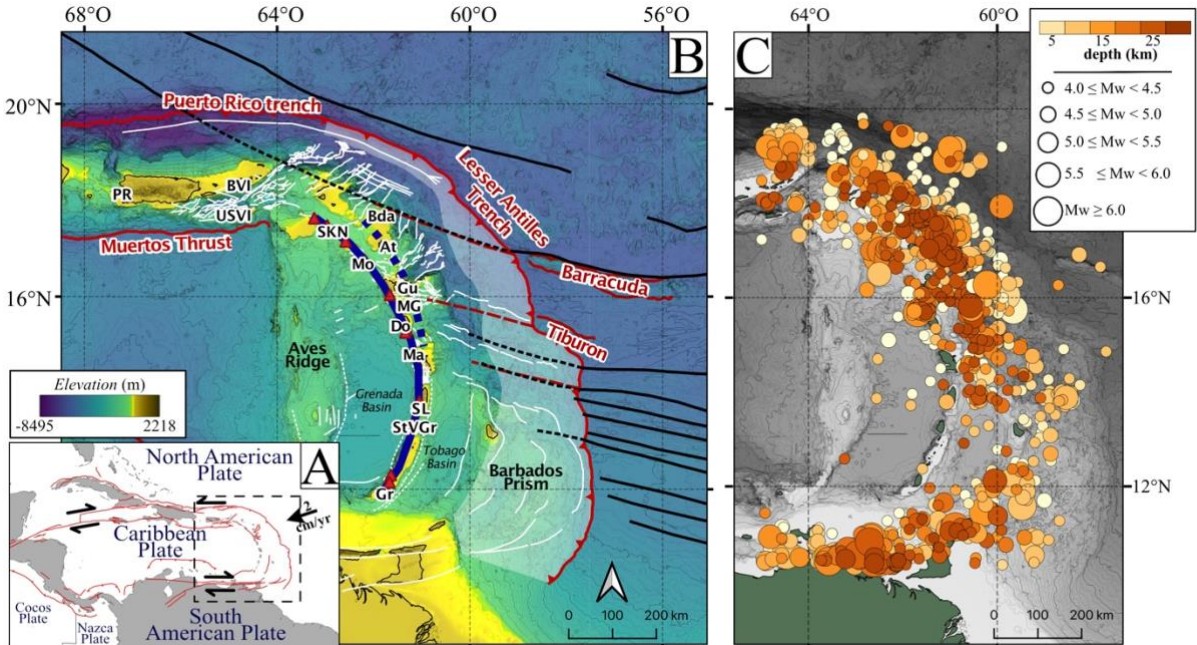

**Figure 1: Seismotectonic framework of the Lesser Antilles. A: Regional tectonic, black arrows: plate motions; red lines: main faults from the Global Earthquake Model; dashed square: B map extent. B: Lesser Antilles structural features; white area: accretionary prism; red triangles: active volcanoes; thick blue solid and dashed lines: active and inactive volcanic arc; black lines: fracture zones; black dashed lines: subducted fracture zone extensions; white lines: known crustal faults; red lines: aseismic ridges; red dashed lines: subducted aseismic ridge extensions; PR: Puerto Rico, BVI: British Virgin Islands, USVI: United States Virgin Islands, Bda: Barbuda, SKN: Saint-Kitts-and-Nevis, At: Antigua, Mo: Montserrat, Gu: Guadeloupe, MG: Marie-Galante, Do: Dominique, Ma: Martinique, SL: Saint-Lucia, StVG: Saint-Vincent-and-Grenadines, Gr: Grenade. Fracture zones are extracted from Global Seafloor Fabric (soest.hawaii.edu) and bathymetry from Global Multi-Resolution Topography (gmrt.org). C: Instrumental seismicity 1906-2021 of the upper plate crust from the ISCU-cat (M≥4).**

The upper plate tectonics are marked by a strong N-S asymmetry. South of ~15° of latitude, the upper plate is characterized by the large Barbados accretionary wedge (e.g., Speed and Larue, 1982; Gomez et al., 2018; Deville, 2023), the Tobago forearc basin, and a single volcanic arc with few major active faults and structures (**Fig. 1B**). In contrast, the northern region is associated with a thin sedimentary wedge, long-term subduction erosion. The forearc basins are affected by trench-perpendicular normal faults, and the active volcanic arc (coupled with an old inactive arc in forearc position) is affected by trench-parallel normal and strike-slip faults (Boucard et al., 2021; Feuillet et al., 2002, 2011b). The N-S asymmetry is also present in the arc crustal thickness increasing from 20–25 km in the south to reach ~35 km in the north beneath St Kitts (Schlaphorst et al., 2018). Across the arc, geodetic velocities indicate very small motions relative to the Caribbean Plate and <1 mm/yr of N-S intra-arc extension (Symithe et al., 2015; van Rijsingen et al., 2021), generally consistent with earthquake focal mechanisms and forearc basin structures (Allen et al., 2019; Lindner et al., 2023).

The overall N-S asymmetry of the Lesser Antilles subduction system also appears in the instrumental seismicity, with higher activity north of ~15° of latitude compared to the southern region (McCann et al., 1984; Hayes et al., 2014; **Fig. 1C**).

However, this pattern does not appear in the historical catalog (Lambert et al., 2009; Bertil et al., 2023; Lambert and Samarcq, 2024; **Fig. 2A**). This discrepancy highlights the difficulty and limits of the seismicity catalogs. Recent damaging earthquakes are associated with magnitudes M=7–7.5 (e.g., 1953 $M_W$=7.3 south of Martinique, 1969 $M_W$=7.2 near Barbados, 1974 $M_W$=7.5 between Barbuda and Antigua, 2007 $M_W$=7.4 north of Martinique, **Fig. 2C**). All are attributed to normal faulting either within the subducting plate (1953, 1969, and 2007, Russo et al., 1992; Dorel, 1981; Régnier et al., 2013) or within the upper plate or mantle wedge (1974, McCann et al., 1982; Feuillet et al., 2002, 2011b). One of the outstanding characteristics of the Lesser Antilles subduction is the lack of instrumental large (M>6.5) thrust earthquake associated with the subduction interface.

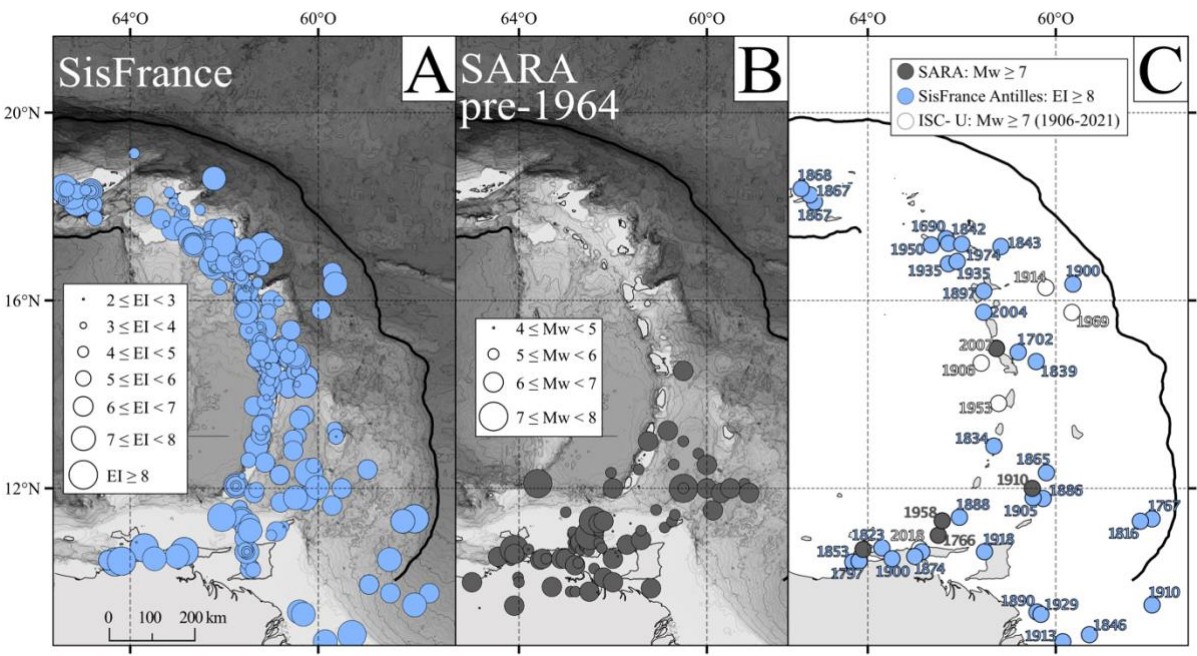

**Figure 2: Lesser Antilles historical seismicity and major earthquakes. Thick black solid line: Lesser-Antilles, Puerto-Rico and Muertos trenches. A: SisFrance historical seismicity. B: SARA pre-1964 historical seismicity C: major instrumental and historical earthquakes from ISCU-cat, SARA and SisFrance catalogs.**

### 3 Method and data

In seismic hazard analysis, seismotectonic zoning models are designed to fill knowledge gaps, and the Lesser Antilles arc and its active seismicity are poorly known. Area sources are built by crossing both qualitative and quantitative criteria with subjective interpretation. When knowledge is sufficient (data collected homogeneously over space and time), the zoning can be purely based on quantitative criteria. In this study, the most recent data has been collected and integrated. The seismotectonic zoning model comprises area sources and fault sources, with the former defining domains of uniform tectonic and seismicity characteristics. Area source boundaries are constructed following three principles, considering existing

knowledge and uncertainties: 1) Boundaries are defined in priority by (a) the seismicity distribution, (b) faults and local tectonics, (c) geodetic data, (d) local geology. 2) A single large zone is preferred to several small zones, unless data clearly shows different tectonic and seismicity characteristics that require zone divisions. 3) Area sources are chosen to prevent seismicity dilution within too large zone. Seismicity distribution, deformation style extracted from focal mechanisms, crustal fault locations and tectonic features were superimposed using Geographic Information Systems (GIS) tool as illustrated in **Fig. 4**, **7**, **8** and **9**. Boundaries depict a consensus between seismotectonic data and discussions with Lesser Antilles experts (Lesser Antilles Working Group). To address geological and tectonic complexities and specificities of the region, we have opted for a division into several zones instead of a simple north-south division. Specific detailed subdivisions are limited to cases where they are supported by the available data. This methodology better constraints each source and allows us to manage uncertainties more precisely, by limiting the propagation of errors to the whole region.

The primary seismicity catalog is a homogenized $M_W$ instrumental catalog (hereafter ISCU-cat) composed of seismicity extracted from the International Seismological Center (ISC, 2023) catalog and regional catalogs from 1906 to 2021 (Bertil et al., 2022, 2023; Bertil, 2024). The $M_W$ homogenization was done using reference magnitudes given by the Global Centroid Moment Tensor (GCMT, Dziewonski et al., 1981; Ekström et al., 2012) project and the National Earthquake Information Center (NEIC, Guy et al., 2015). This catalog is considered complete since 1964 for $M_W \geq 4.3$ and completeness for $M_W \geq 4$ is not reached until 1985 (Bertil et al., 2022, 2023; Bertil, 2024). Taking $M_W \geq 4.0$ for the whole catalog limits bias in regional completeness, even if few data exist before 1985 (6%). No hypocenter relocations were conducted but a first order quality score is supplied based on phase number (pn) from A (pn $\geq$ 1000) to E (pn $\leq$ 3) instead of location uncertainties. The ISCU-cat comprises 0.2% A, 5.3% B, 18.6% C, 75.7% D and 0.2% E. The ISCU-cat spatial variations in seismicity are analyzed to determine area source boundaries and their associated Gutenberg-Richter distributions. The ISCU-cat needs for magnitude estimation improvements are discussed in section **5**.

The ISCU-Cat is completed for local analysis around Guadeloupe and Martinique with the instrumental catalogs IPGP (Globe Physic Institute of Paris, Saurel et al., 2022) and CDSA (seismological data center for the French lesser Antilles, Massin et al., 2021) and regionally by the historical catalogs SisFrance Antilles (Vermeersch et al., 2002; Lambert et al., 2009; Bertil et al., 2022; Lambert and Samarcq, 2024) and SARA (Gómez-Capera et al., 2017) (**Fig. 2A, 2B**). SisFrance comprises 19% of quality B (corresponding to ~10 km location uncertainty) and 81% of quality C (10-20 km uncertainty). For the SARA pre-1964 catalog, no uncertainties or quality are available. Historical seismicity allows us to discuss past activity of specific regions and to compare it with instrumental seismicity and geodetic data (Foix et al., 2023a). The exact location of some historical events is still debated, such as the 1839 and 1843 earthquakes. Bernard & Lambert (1988) and McCann et al. (1984) interpreted them as megathrust earthquakes whereas van Rijsingen et al. (2021) proposed that the 1843 event had a smaller magnitude, or different mechanism or location within the subducted slab, and that the 1839 event could also be located in the subducted slab.

We construct a composite catalog of earthquake focal mechanisms comprising 572 events from the GMCT (Dziewonski et al., 1981; Ekström et al., 2012), ISC (Letas, 2018; Letas et al., 2019) as well as Corbeau et al. (2019, 2021), González et al. (2017), and Ruiz et al. (2013), from 1977 to 2021, hereafter named as FMAnt2021 (Focal Mechanisms Antilles 2021). From FMAnt2021, we compute average faulting types and P and T axis orientations on a regular grid using Mazzotti et al. (2021) method (smoothing distance of 40 km, minimum of 3 mechanisms within a radius of 50 km). The FMAnt2021 allow us to determine area source boundaries and deformation types.

The oceanic subducted plate surface is based on a unification of the slab models of Bie et al., (2019) and Laurencin et al., (2018) (**Fig. 3B**, **S2**). We have georeferenced and digitized these interface geometries to combine them in one unique surface using a GIS tool every 10 km of depth. We then transformed it into a grid that can be extrapolated to get a surface and to be used for earthquake sorting. Paulatto et al. (2017) slab and Moho geometries were not used in order to keep consistency along the arc. The slab top geometry may vary according to the publications, but also according to the interpretation of the presence or absence of a slab. Beneath the Lesser Antilles central area, at depths of around 170-200 km, Lindner et al. (2023) observed a seismic gap in the lithosphere of subducted American plates, whereas Braszus et al. (2021) observed a continuous slab, in agreement with the tomography of Bie et al. (2019). These differences would have an impact on earthquake sorting and the resulting statistics.

No unified database of crustal faults exists for the Lesser Antilles, and knowledge is heterogeneous along the arc. We use crustal faults from the Global Earthquake Model of the Caribbean region (Styron et al., 2020) completed with local studies of Boucard et al. (2021), Feuillet et al. (2001, 2002, 2004, 2011a, 2011b), Garrocq et al. (2021), Laurencin et al. (2017, 2019), and Leclerc et al. (2016) (**Fig. 1B**). Various unknowns and interpretations remain on fault activities. The Anegada passage fault system (**Fig. 3 (5)**) motion was interpreted from extensional faulting to sinistral or dextral transtension (Laurencin et al., 2017 and references therein). Fault activity is sometimes debated, such as for the V-shaped basin faults from Guadeloupe to Saint-Kitts-and-Nevis (Feuillet et al., 2001, 2011a; Boucard et al., 2021). Moreover, structures still need to be imaged and understood south of Saint-Lucia. Finally, we use the geodetic velocities from van Rijsingen et al. (2021) to calculate geodetic strain rates (**Fig. 4**) and, completed by micro-atoll subsidence data (Philibosian et al., 2022), to test models of megathrust interseismic coupling on 2D cross-sections. We also use the geodetic velocities to calculate extension rate for some specific regions and determined an associated standard uncertainty ($\sigma/\sqrt{N}$; σ: standard deviation, N: measurements).

Details of the area source limits and geometries are given in the supplementary material (**S1**, **S2** and **S3**) and in the *Risk Prevention Department* report (Foix et al., 2023a) with information on the Gutenberg-Richter distribution for each area

source (**S4**). Supplementary Material for this article includes also minimum magnitude sensitivity analysis impact for the Marie-Galante graben (**S5**) and a comparison between Martin and Combes (2001) and this study (**S6**).

## 4 Seismotectonic zoning model

### 4.1 Upper plate seismotectonics, area sources and faults

#### 4.1.1 Upper plate seismotectonics

At its northern end, the Lesser Antilles subduction is limited by the Anegada Passage (**Fig. 3A**), an E-W *en-échelon* strike-slip system (Laurencin et al., 2017) with low relative motion based on geodetic data (Symithe et al., 2015), terminating eastward in the pull-apart Sombrero Basin (Laurencin et al., 2019). The Bunce - Bowin fault sinistral strike-slip system marks the limit between the thin accretionary wedge and the upper plate backstop (**Fig. 3A**). Its maximum slip rate is ~16 mm/year (Laurencin et al., 2019), with no known large earthquake on this fault. Its small depth extent (~5 km) suggests that only moderate earthquakes can be expected (ten Brink and Lin, 2004).

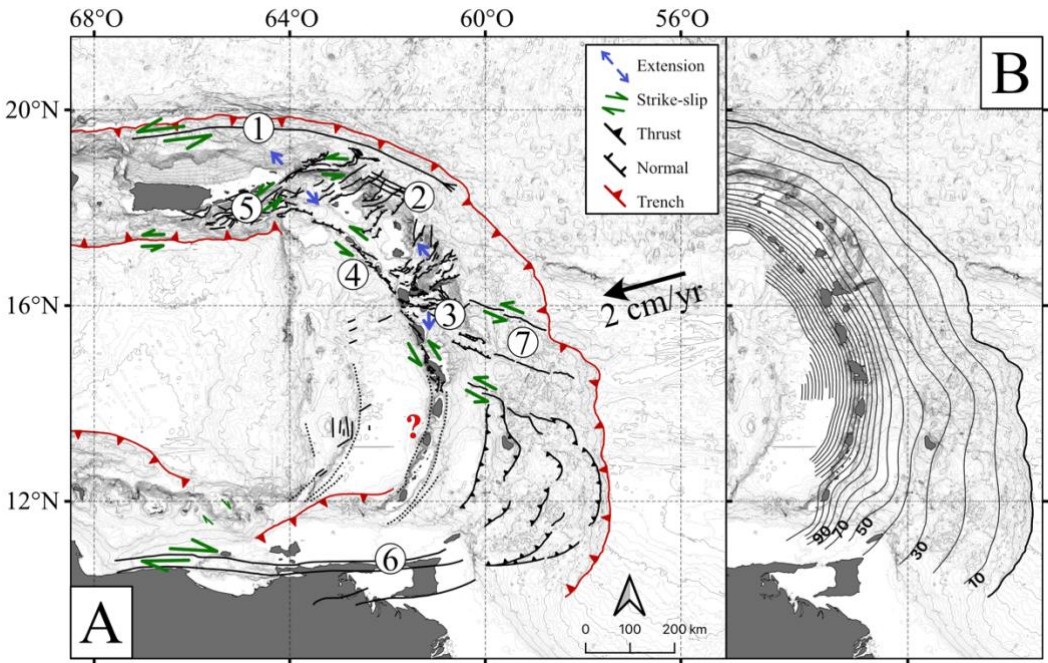

**Figure 3: Crustal faults and slab geometry. A: crustal faults (black solid lines) and relative motions (blue and green arrows), cf. sect. 4.1; red question mark: lack of knowledge on possible normal faults along the arc; (1) Bunce Fault, (2) Tintamarre Faults crosscutting a V-shaped basin, (3) Marie-Galante graben, (4) Bouillante-Montserrat Fault, (5) Anegada Passage Fault system, (6) El Pilar Fault system, (7) Lateral Ramp. B: Bie-Lau-Slab unified slab top geometry, cf. sect. 4.1 and S2.**

The northern half of the Lesser Antilles arc, from Antigua to Guadeloupe, is characterized by Paleogene V-shaped basins (**Fig. 3A**), inferred to be related to the collision of the Caribbean Plate with the Bahamas Bank and partly overlapped by normal faults (e.g., Tintamarre Fault, **Fig. 3A**) attributed to Mid-Miocene margin erosion (Boucard et al., 2021). The V-shaped structures are interpreted to be inactive in the north (Symithe et al., 2015; Boucard et al., 2021) and active east of Guadeloupe (Feuillet et al., 2001). Present-day tectonics is characterized by trench-perpendicular normal faults bounding grabens and spur in the forearc region, such as the Marie-Galante graben (**Fig. 3A**) or the Bertrand-Falmouth spur, associated with N-S to NNW-SSE extension (Feuillet et al., 2001, 2002). Along the active volcanic arc, trench-parallel *en-échelon* faults accommodate sinistral motion, such as the Bouillante-Montserrat fault system (**Fig. 3A**), or normal motion, such as the Roseau Fault (Feuillet et al., 2001, 2011a; Leclerc and Feuillet, 2019). Shallow seismicity attests of normal and strike-slip faulting on these trench-parallel and -perpendicular structures (Lindner et al., 2023), e.g., the 2004 $M_W$=6.3 Les Saintes earthquake on the Roseau Fault (Bazin et al., 2010; Feuillet et al., 2011b; Escartín et al., 2016; **Fig. 2C**). These recent fault systems inferred to be related to strain partitioning of the oblique plate convergence (Feuillet et al., 2001, 2011a).

The transition with the southern Lesser Antilles region is marked by lateral ramps following the Barracuda and Tiburon ridges (Brown and Westbrook, 1987; **Fig. 3A**). South of Saint-Lucia, the fault system knowledge is sparse and a possible extension of the northern trench-parallel *en-échelon* normal faults remains unknown. Large normal faults have been interpreted along both sides of the volcanic arc from south of Martinique to Grenade (Christeson et al., 2008; Aitken et al., 2011; **Fig. 3A question mark**). However, recent seismic reflection data show no clear evidence of such large faults (Garrocq et al., 2021). They only indicate normal faults along the west flank of the arc, between Saint-Vincent-and-the-Grenadines and Saint-Lucia, possibly similar to those north of Martinique.

### 4.1.2 Upper plate area sources

The primary area source division is trench-parallel, to take into account the differences in structure, tectonics, and seismicity between the accretionary wedge (AW), forearc (FA), arc (A), and back-arc (BA) regions of the upper plate (**Fig. 4A**). We also include in the upper-plate area sources seismicity in the outer rise of the downgoing plate (DP). The secondary division is trench-perpendicular based on structures, fault types, and seismic activity in the different regions. The bottom limits of these area sources are based on the seismicity characteristics in each zone or on the structural limits. We fix the bottom limit of the crustal upper-plate area sources either at the Moho (set at 28 km, Kopp et al., 2011; Bie et al., 2019) or at the downgoing plate surface depth (minus 5 km, to avoid interface seismicity). The diffuse distribution of the crustal seismicity, due to the network geometry, does not allow a more specific definition. An alternative for the crustal source areas is to set their bottom limit at an assumed seismogenic depth of 15-20 km, similar to that observed for active faults. The bottom limit of crustal downgoing-plate area sources is fixed at 10-15 km depth. Due to low seismicity records and high distance of these

sources to the islands, the choice of this depth has no impact on hazard calculations. In the following, we highlight the main points of interest and sources of uncertainties:

- DP (downgoing plate) sources: The seismotectonics of these zones are poorly constrained, with very limited seismicity and only two moderate instrumental events (2003 M=6.6 and 2016 M=5.6). Large potentially seismogenic structures are observed down to 10-15 km depth on the oceanic plate outer rise (Marcaillou et al., 2021; Allen et al., 2022), that could generate large normal or strike-slip earthquakes. Based on global analogs, outer rise seismicity may reach magnitudes M~8–8.5 (e.g., Sumatra, Japan, Meng et al., 2012; Kanamori, 1971).

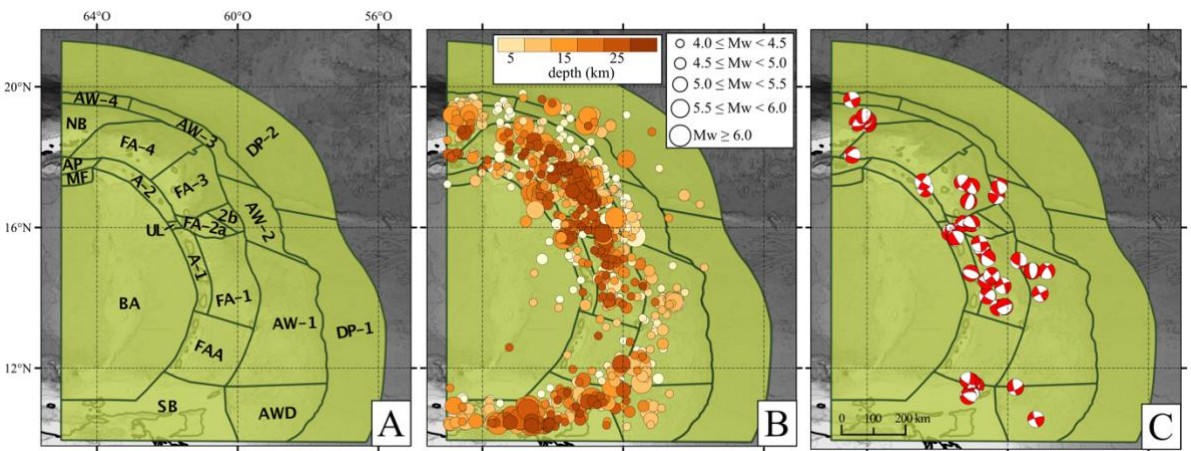

**Figure 4: Lesser Antilles upper-plate crust seismotectonic zoning model. A: crustal area sources; SB: South Boundary, AW: Accretionary Wedge, AWD: AW Death, FA: Forearc, A: Arc, BA: Backarc, MF: Muertos Fault, NB: North Boundary, AP: Anegada Passage, UL: Undefined Limit, DP: Downgoing Plate. B: crustal seismicity from ISCU-cat (M ≥ 4), Bertil, (2024). C: crustal focal mechanisms from FMAnt-2021.**

- AW (accretionary wedge) sources: These zones are divided along strike to account for the evolution from the large Barbados prism in the south (AW-1) to the narrow prism under erosion in the north (AW-3 and 4). The transition (AW-2) is defined by the Tiburon and Barracuda Ridge extensions beneath the wedge (**Fig. 1B**). The southern end corresponds to a zone of progressive termination (AWD), where the accretionary wedge gives way to more compacted material. This boundary is not clearly defined. The seismogenic potential of these sources is poorly constrained due to the low precisions of earthquake locations. A few moderate events may be associated with these zones without certainty (e.g., 1922 M=6.1, Russo et al., (1992), 2015 M=5.7 clusters, 1767 and 1816 historic earthquakes, **Fig. 2C**, Le Roy et al., (2017)). Ocean Bottom Seismometer (OBS) recordings show no seismicity over a six-month period in the wedge off Guadeloupe (Laigle et al., 2007, 2013; Ruiz et al., 2013). Worldwide, accretionary prisms are associated with high fluid content, unconsolidated sediment, and low-frequency earthquakes or tremors (Ito and Obara, 2006; Obana and Kodaira, 2009). They are not associated with really well-known earthquakes but very few standard events occurred (e.g., 2023 M=5.5 Panama, Bradley and Hubbard, 2023).

- Forearc (FA) and arc (A) sources: South of Saint-Lucia, the forearc and arc are grouped in a common area source (FAA) due to the low level of instrumental seismicity and lack of identified major fault. The northern limit corresponds to a smooth northward increase of seismicity. The geodetic velocity increase between Saint-Lucia and Saint-Vincent-and-the-Grenadines is equal to $0.6 \pm 0.3$ mm/yr and $1.4 \pm 0.4$ mm/yr toward the south, for the eastern and northern components, in the Caribbean Plate reference frame (**Fig. 5**). This is not associated with focal mechanisms marking extension (**Fig. 4C**). The central and northern regions are characterized by a distinction between arc (A) zones, associated with arc-parallel normal and strike-slip faults, and forearc (FA) zones, associated with NW-SE *en-échelon* faults (**Fig. 1B, 3A**). The lateral divisions correspond to changes in instrumental seismicity rates or to independent seismotectonic features such as the Marie-Galante graben (FA-2). The westward extension of the Marie-Galante graben is uncertain, with two alternative sources (FA-2a or FA-2b). Similarly, the transition between A1 (normal faulting) and A2 (normal to sinistral faulting) is uncertain and associated with the alternative limits UL. The seismogenic potential of the forearc and arc is characterized by seismicity clusters with a more sustained activity north of Saint-Lucia (**Fig. 4B**). Large events associated with arc tectonics are the 2004 $M_W$=6.3 Les Saintes and the 1867 $M_W$=7.2 Virgin Islands earthquakes, whereas the 1967 $M_W$=6.4 earthquake struck the forearc (**Fig. 2C**). Larger historical events may have occurred in the arc and forearc (1690 $M_S$=7–8 near Barbuda, 1839 (M=7.5-8) and 1843 (M=8-8.5) offshore Martinique and Guadeloupe, 1867 $M_S$=7.2 near the Virgin Islands), but their exact locations and associated structures are unknown. The 1843 moderate tsunami intensity (Lambert and Terrier, 2011) and the possible 1839 sea disturbance (Clouard et al., 2017) favor intraslab or deep interface locations for these two earthquakes.

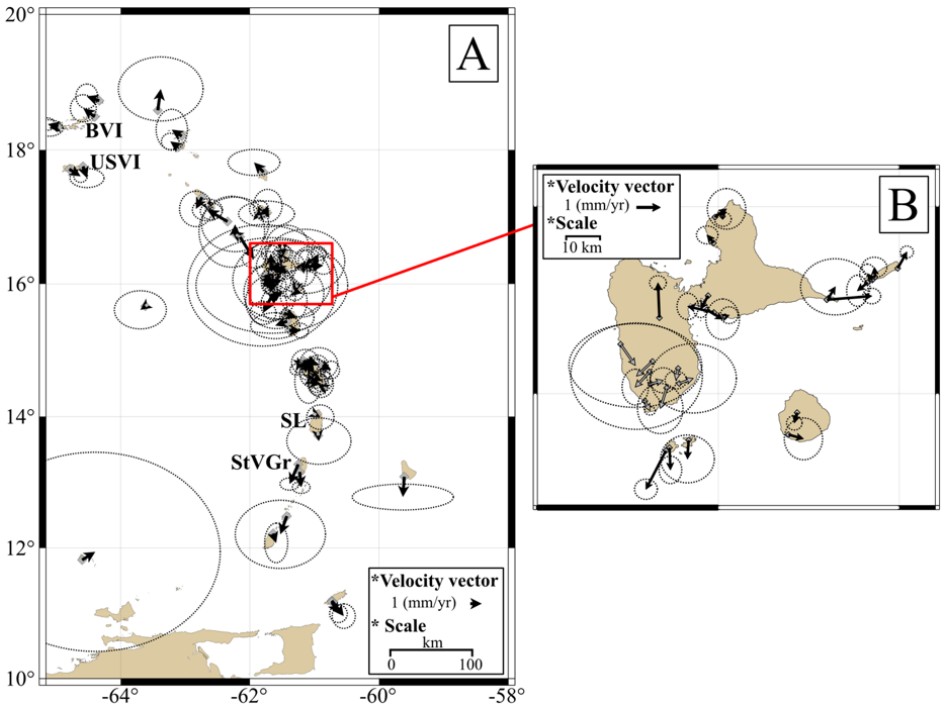

**Figure 5: Geodetic velocities of the Lesser Antilles arc in the Caribbean Plate reference frame (van Rijsingen et al., 2021). Black arrows: velocity vectors; black dashed circles: error ellipses with 95% confidence. A: Lesser Antilles arc. B: zoom on Guadeloupe; Grey arrows: velocity vectors related to La Soufrière volcano.**

• Boundary zones are defined at the northern (NB), southern (SB), and western back-arc (BA) limits of the model to capture diffuse seismicity without specific information. Two smaller zones are defined along the southern Anegada Passage (AP) and the Muertos Fault (MF). The AP zone is associated with few M≥4 instrumental earthquakes (**Fig. 4B**), the historical 1867 $M_W$=7.2 Virgin Island earthquake (**Fig. 2C**), and a geodetic velocity gradient of $1.8 \pm 0.4$ mm/yr and $-0.55 \pm 0.2$ mm/yr for the eastern and northern components (**Fig. 5**), highlighting the uncertainty on local active
tectonics.

### 4.1.3 Upper plate fault sources

Fault geometries and slip rates are required to integrate fault sources in seismic hazard models. Only a few Lesser Antilles faults meet these criterions: the Roseau, Morne Piton, and Bouillante-Montserrat Faults near Guadeloupe, the Redonda Fault between Saint-Kitts-and-Nevis and Montserrat, and the Anegada Passage and Muertos Trough at the northern end (**Fig. 6A**).
Slip rates and maximum structural magnitudes based on Wells and Coppersmith (1994) scaling law are given in (**Table 1**). Additional faults and structures have been studied on land and offshore, such as the May Fault (Sedan and Terrier, 2001) or the North Lamentin and Schoelcher Faults (Terrier, 1996), but the available data are not considered robust enough to be included in a seismic hazard model at this time.

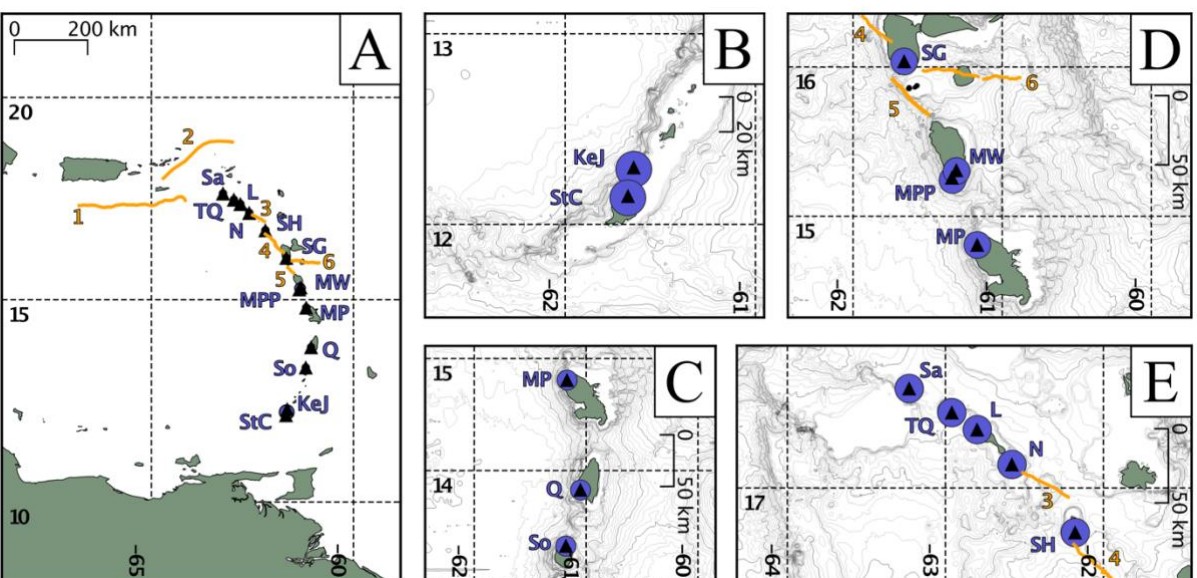

**Figure 6: Lesser Antilles volcanic (purple circles) and fault (orange solid lines) seismogenic sources. A: Geographical repartition of active volcanoes and source faults; black triangles: active volcanoes; StC: Sainte Catherine, KeJ: Kick'em Jenny, So: Soufrière of Saint Vincent, Q: Qualibou, MP: Mount Pelée, MPP: Morne Plat Pays, MW: Morne Watt, SG: Soufrière of Guadeloupe, SH: Soufrière Hills, N: Nevis, L: Liamuiga, TQ: The Quill, Sa: Saba; 1: Muertos Trench, 2: Anegada Fault system, 3: Redonda Fault,**

| Faults | Length (km) | Seismogenic Depth (km) | Slip Rate (mm/yr) | Mmax.struct. | Mmax.obs | References |
|---|---|---|---|---|---|---|
| Roseau | 35 | 15 | 0.15 to 0.4 | 7.0 | Mw=6.3 (2004) | Feuillet, Beauducel, Jacques, et al., (2011); Leclerc et al., (2016) |
| Morne Piton | 50 | 20 | 0.20 ± 0.05 0.5 ± 0.2 | 7.5 | Ms~5.5 (1851) | Philippon et al. (2024); Feuillet et al. (2004) |
| Redonda | ~30 | 15? | ~0.2 | 7.0 | Ms=6.2 (1985, z=9±2km) | Carey et al., (2019): 0.3 mm/yr of regional subsidence with 0.16 mm/yr, 60° of fault dip is assuming. Feuillet et al., (2010) fig. 2. |
| Bouillante-Montserrat | 10 to 20 km segments (~60) | 15? | ~0.3 0.15-0.20 | 7.3 | ? | Beck et al., (2012): 10 m in 3500 year; Philippon et al. (2024); Feuillet et al., (2010) |
| Anegada | 220 | ? | 1.0 to 1.7 | 6.8-8.0 | Ms=7.5 (1867) | Zimmerman et al., (2022) and references therein |
| Muertos Trough | 641 | - | 0.7 | 7.6 (East segment) | Ms=6.7 (1984) | Zimmerman et al., (2022) and references therein |

**Table 1: Seismogenic fault sources characteristics of the fault's seismogenic source.**

### 4.1.4 Marie-Galante Graben

The Marie-Galante graben corresponds to the FA-2a seismotectonic zone (**Fig. 4A**). Guadeloupe is defined as a key
transition area between the northern and southern Lesser Antilles arc in terms of geodetic velocities, seismicity rates, fault
orientations and tectonic style. Arc-parallel (Roseau, Bouillante-Montserrat) and arc-perpendicular (Marie-Galante graben)
faults intersect near the Soufrière volcano. The northern part of Guadeloupe is marked by northward GNSS velocities, while
the southern part is characterized by southward velocities (**Fig. 5**). The graben may have generated three historical
earthquakes in the last 150 years, with intensities from VII to VIII, including the destructive April 29, 1897, M=5.5–6
earthquake located closed to Pointe-à-Pitre (**Fig. 2C**) and linked with the Gosier Fault (Bernard and Lambert, 1988). The
Morne Piton Fault (**Fig. 6A, number 6**) may be responsible for the 1851 earthquake (EI=VII; Feuillet et al. 2011b). The
structural characteristics of the Morne Piton Fault are compatible with potential M=6.5 earthquakes every 1400 to 3300
years, or M=5.5 every 400 to 1000 years (Feuillet et al., 2004). The Morne Piton Fault slip rate is estimated between 0.2 ±
0.05 mm/yr and 0.5 ± 0.2 mm/yr (Philippon et al., 2024; Feuillet et al., 2004).

We estimate the graben overall extension rate from the ISCU-cat and geodetic data. From the difference in geodetic velocities between northern and southern Guadeloupe, the extension rate is ~1.1 ± 0.2 mm/yr for the northern component. A formal strain rate inversion of these velocities yields an extension rate of ~0.6 mm/yr integrated over a distance of 28 km. For the seismicity data (ISCU-Cat), we use the 743 events in FA-2a to estimate a Gutenberg-Richter magnitude frequency distribution and its associated seismic moment and deformation rate (**S4**, Mazzotti and Adams, 2005). Using minimum and maximum magnitudes of 3.0 and 7.3 (based on fault geometry), a seismogenic thickness between 15 and 20 km and a normal fault geometry (length=135 km, with=75 km, dip=60° and rake=90°), we estimate an extension rate from 0.5 to 1 mm/yr (with a b-value of 1.2-1.5 – **S5**). These estimated graben extension rates of 0.5-1.1 mm/yr are compatible with the Morne Piton Fault slip rate (0.2-0.5 mm/yr). The slip rates of other graben faults have not been estimated yet and may accommodate a part of the deformation.

## 4.2 Subduction interface seismotectonics and area sources

### 4.2.1 Subduction interface seismotectonics

The seismogenic potential of the Lesser Antilles megathrust is a major conundrum. No large megathrust earthquake has been recorded in the instrumental period, while the historical 1839 (M=7.5-8) and 1843 (M=8-8.5) earthquakes may be associated with the interface, the subducted slab, or the upper plate with higher probabilities for deep interface or intraslab (Bernard and Lambert, 1988; Feuillet et al., 2011a; Hough, 2013). North of Guadeloupe, the interface seismicity is highlighted by moderate M=4.5–6.5 reverse-faulting earthquakes down to 60–65 km depth (Lindner et al., 2023), whereas very few similar events occur to the south (**Fig. 7B, 7C**). In contrast, no interface earthquakes were recorded below 20 km depth during a 6-month OBS deployment offshore Guadeloupe and Martinique (Laigle et al., 2013a). The 60–65 km downdip extent is consistent with thermal modeling of the subduction system, which shows that the seismic-aseismic transition ~350 °C occurs ~65 km below Martinique and Saint-Martin (Ezenwaka et al., 2022), with its along strike variations are compatible with the interface seismicity distribution (Gutscher et al., 2013).

Geodetic data indicate that the present-day interseismic coupling of the whole megathrust is very low (~10%) (Symithe et al., 2015; van Rijsingen et al., 2021). Coral and micro-atoll subsidence rates are interpreted as indicative of strong interseismic coupling on the megathrust at ~40–80 km depths between Martinique and Barbuda (Weil-Accardo et al., 2016; Philibosian et al., 2022). van Rijsingen et al. (2022) observed subsidence from geodetic vertical motions, in coherency with coral and micro-atoll data and modelled that a locked or partially-locked interface would produce uplift. In order to test the compatibility of Philibosian et al. (2022) and van Rijsingen et al. (2021) results, we performed a first-order simulation of the megathrust interface interseismic coupling depth along a 2D cross-section (**Fig. 8**). We did not consider the 3D slab geometry. We simulated coupling depth proposed by Philibosian et al. (2022) independently to our proposed area sources. Results from van Rijsingen et al. (2021, 2022) as well as simple 2D cross-section models (**Fig. 8**) show that the shallow 0–40

km-depth section of the megathrust must be associated with very low coupling to be compatible with horizontal geodetic velocities. Vertical velocities indicate a general subsidence rate that could, in part, be associated with strong deep (40–70 km) interseismic coupling, although details of this conclusion are debated (Philibosian et al., 2022; van Rijsingen et al., 2022).

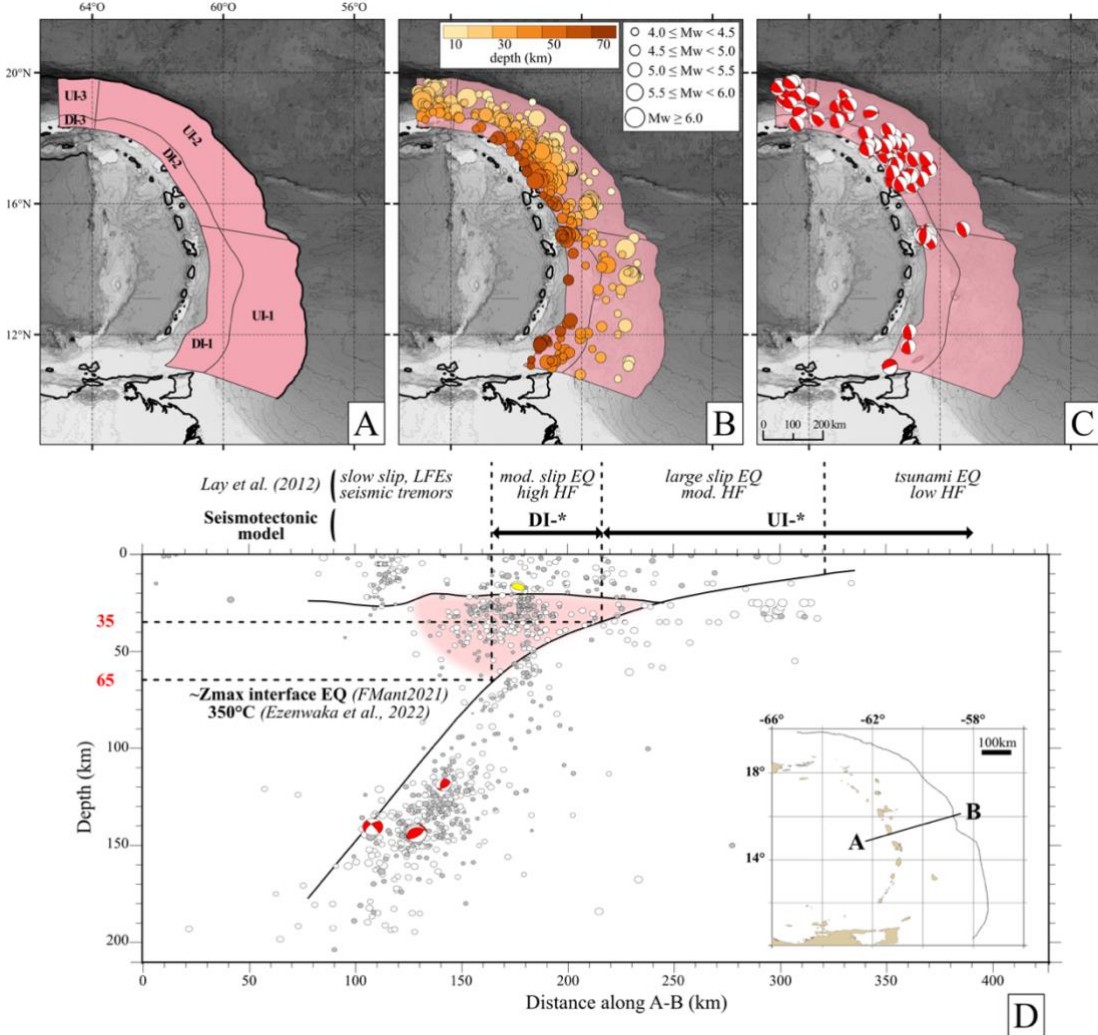

**Figure 7: Lesser Antilles interface seismotectonic zoning model. A: interface area sources; UI-\*: Upper-Interface, DI-\*: Deep Interface. B, C: same as Fig. 5 but for the interface seismicity. D: interface conceptual model based on Lay et al. (2012) and adapted for the Lesser Antilles; AB cross-section, reported in the inset with corresponding domain (mod.: moderate; HF: High Frequency; LFEs: Low Frequency Events; EQ: Earthquakes); seismicity from the ISCU-cat (white dotes) and CDSA (grey dotes) along AB, thick black solid lines: slab and Moho from Paulatto et al., (2017), focal mechanisms from FMAnt-2021, thin black dashed lines: domain depth limits, IU-\* bottom limit is marked at 35 km depth and DI-\* bottom limit at 65 km depth, pink zone: mantle wedge seismicity.**

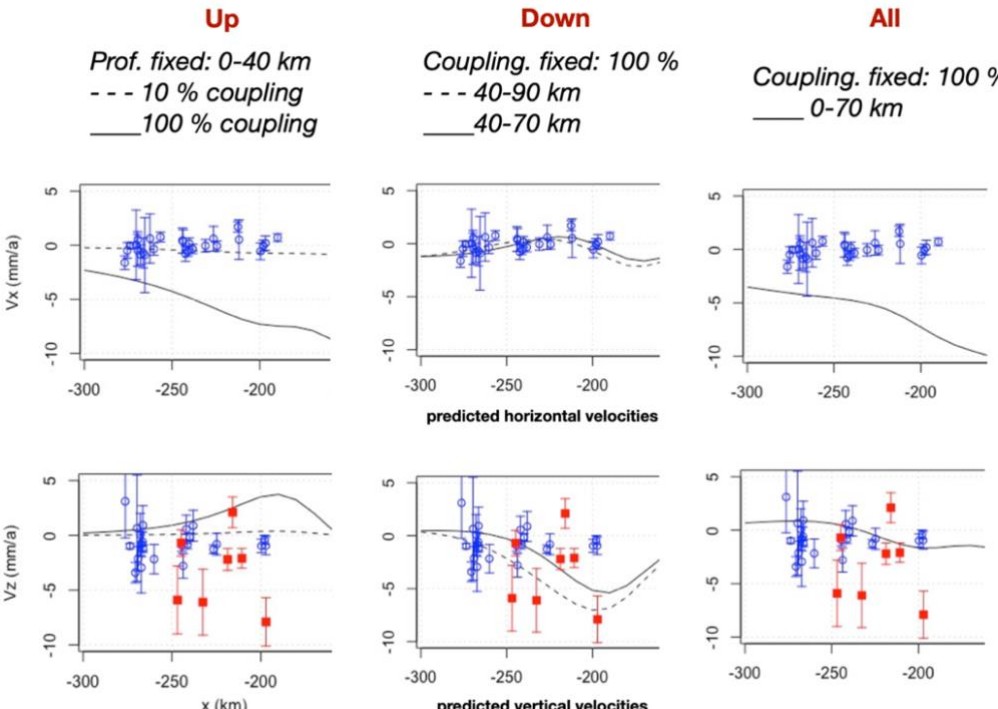

**Figure 8: Lesser Vertical (Vz) and horizontal (Vx, in the profile direction) velocities from GNSS (blue square) and micro-atoll (red square) data compared to predictions from 2D models of subduction interface interseismic coupling for a cross section at the latitude of Guadeloupe. Three cases are considered: interseismic coupling of the upper part of the interface (Up, 0-40 km depth), interseismic coupling of the deep part (Down, 40-90 km depth), and interseismic coupling of the entire interface (All). Micro-atoll and GNSS data are from Philibosian et al. (2022) and van Rijsingen et al. (2021).**

### 4.2.2 Subduction interface area sources

In order to account for this conflicting information, we propose a seismotectonic zoning model of the Lesser Antilles megathrust based on the reference model of Lay et al. (2012), which divides the subduction interface into four downdip domains based on seismic behavior and high-frequency radiations (HF): 0–15 km depth = tsunami earthquakes and low HF, 15–35 km depth = great earthquakes and moderate HF, 35–55 km depth = large earthquakes and strong HF, >55 km depth = stable slip, slow-slip events, very-low-frequency earthquakes. A previous division of the subduction interface for tsunami hazard was proposed based on plane dipping (IOC-UNESCO, 2020). Our simplified version of this model adapted to seismic hazard assessment for the Lesser Antilles corresponds to (**Fig. 7A, 7D**):

- An upper interface (UI) from 0 to 35 km depth capable of generating great M=8–9 megathrust earthquakes but associated with very low interseismic coupling. Assuming an average slip per event of 5–10 m, an interseismic coupling

of 10%, and a convergence rate of 20 mm/yr, the great earthquake return period is ~2500–5000 years compatible with sedimentary records (Seibert et al., 2024).

- A deep interface (DI) from 35 to 65 km depth that can generate large M=7–8 earthquakes and associated with either low
interseismic coupling (model 1, compatible with geodetic data) or high interseismic coupling (model 2, compatible with coral data). We propose a lower weight on Model 2 because of the high variability and large uncertainties of the coral data.

The depth limits between UI and DI are assumed constant. An along-strike division between area sources UI-1 / DI-1 and
UI-2 / DI-2 is set near the Barracuda and Tiburon Ridges depth extensions (**Fig. 1B, 7A**) to reflect the North / South American Plate boundary and the N-S differences in oceanic plate fracturing, megathrust seismicity, and convergence direction. The issue of lateral earthquake propagation across this lateral limit is unresolved. In addition, minor area sources UI-3 and DI-3 defined the eastern end of the Puerto Rico subduction zone.

### 4.3 Subducted slab seismotectonics and area sources

Seismotectonics of the subducted oceanic plate is characterized by high seismic activity down to ~150–200 km depth (**Fig. 9B**) associated with downdip extension, normal, and strike-slip faulting (Gonzalez et al., 2017; Lindner et al., 2023; **Fig. 9C**). Intraslab seismicity is particularly abundant below Martinique and Dominique, potentially in relation with the subduction of fracture zones (Bie et al., 2019; Lindner et al., 2023). The 2007 $M_W$=7.4 earthquake occurred in this region and is among the strongest instrumental earthquake recorded in the Lesser Antilles. Larger historical earthquakes, such as the
1839 or 1843 events, may be associated with intraslab seismicity (van Rijsingen et al., 2021).

Intraslab source areas are primarily based on a downdip division constrained by the slab geometry and the seismicity mechanisms (**Fig. 9C**). Secondary along-strike divisions are defined to account for seismicity density and mechanism variations (**Fig. 9A**):
- Shallow slab (SS) sources, from 0 to 30 km depth, correspond to a constant 10–15° slab dip and heterogenous along-strike seismicity distribution.
- Slab bending (SB) sources, from 30 to 80 km depth, correspond to the region of progressive increase in slab dip.
- Slab intermediate (SI) sources, from 80 to 155–190 km depth, are characterized by a constant slab dip ~55° in the north and ~40° in the south. The deeper SI-3 limit (190 km vs. 155 km in other sources) correspond to the extent of high
seismic activity.
- The slab detachment (SD) source, from 155–190 to 280 km depth, covers the maximum possible slab extent based on seismic tomography (Braszus et al., 2021).

The seismogenic characteristics of these sources is illustrated by instrumental and historical earthquakes. The 2014 $M_W$=6.4 earthquake is located in the shallow slab (SS-2), with a normal-faulting rupture under the accretionary prism ~11–15 km depth. The 1969 $M_W$=7.2 normal-fault earthquake can be associated with either the SS-2 or SB-3 due to the uncertainty in its focal depth (Stein et al., 1982). The 2007 $M_W$=7.4 earthquake (oblique normal-faulting, 152 km depth) is associated with the deeper SI-3 source and caused moderate damages in Martinique (Schlupp et al., 2008; Régnier et al., 2013). Apart from the poorly constrained 1969 $M_W$=7.2 event, SB sources do not include instrumental large M>6 earthquakes. Worldwide, the largest intraslab earthquakes can exceeded M~8 (e.g., 1939 Ms=7.8 south central Chile, Beck et al. 1998; 2017 $M_W$=8.2 Mexico, Jiménez, 2018).

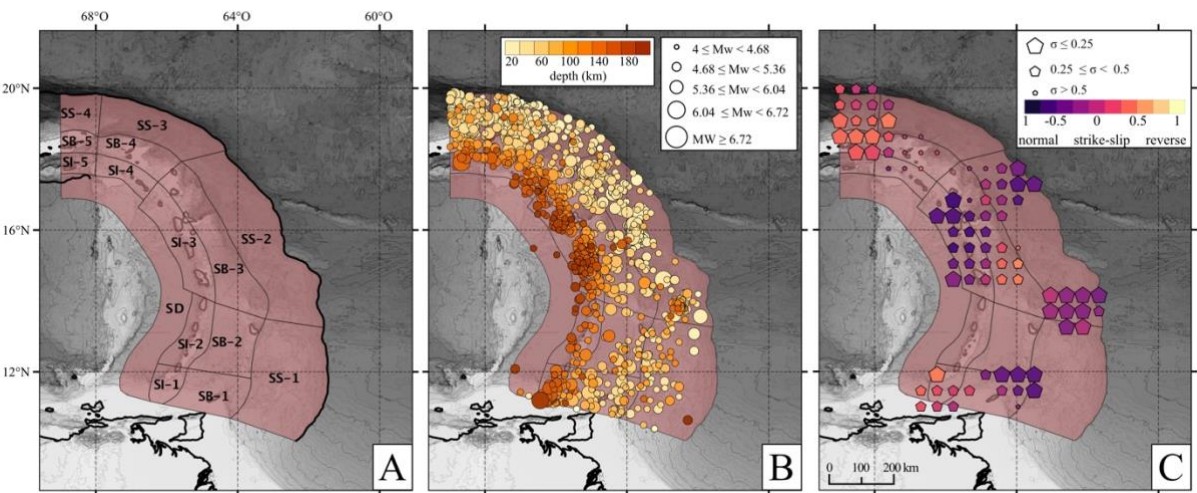

**Figure 9: Lesser Antilles intraslab seismotectonic zoning model. A, and B: same as Fig. 5 but for intraslab seismicity; SS: Shallow Slab, SB: Slab Bending, SI: Slab Intermediate, SD: Slab Detachment. C: grid-average faulting style based on instraslab FMAnt-2021 events, with symbol sizes inversely proportional to the standard deviation.**

## 4.4 Mantle wedge seismotectonics and area sources

The Lesser Antilles subduction is characterized by a cold mantle wedge associated with normal-faulting seismicity (Bie et al., 2019, 2022; Laigle et al., 2013b; Ruiz et al., 2013), as also observed in the Greek, New-Zealand, and Northern Japan subductions (Uchida et al., 2010; Davey and Ristau, 2011; Halpaap et al., 2019, 2021). This peculiar "supra-slab" seismicity may be explained by the presence of pyroxenitic material within peridotites instead of aseismic serpentinized peridotite (Laigle et al., 2013b), or by subduction fluids from slab source expelled to the mantle wedge (Haalpap et al., 2019) and which could result in a cold mantle wedge (Hicks et al., 2023).

The mantle wedge seismicity is located ~50 km east from the volcanic arc and between 25 and 60 km depth (**Fig. 7D**). Using the subducting slab and arc Moho geometries as limits, we identify over 3000 events associated with mantle wedge seismicity (**Fig. 10B**), allowing us to define four area sources where the bottom limit follows the slab surface minus 5 km down to 70 km depth, and the top limit is limited by the crustal Moho at 28 km (**Fig. 10A**): A main mantle source (MS) from

Saint-Lucia to Barbuda characterized by high seismic activity, two southern and northern mantle sources (SMS, NMS) with lower seismic activity, and a Puerto Rico mantle source (PRM).

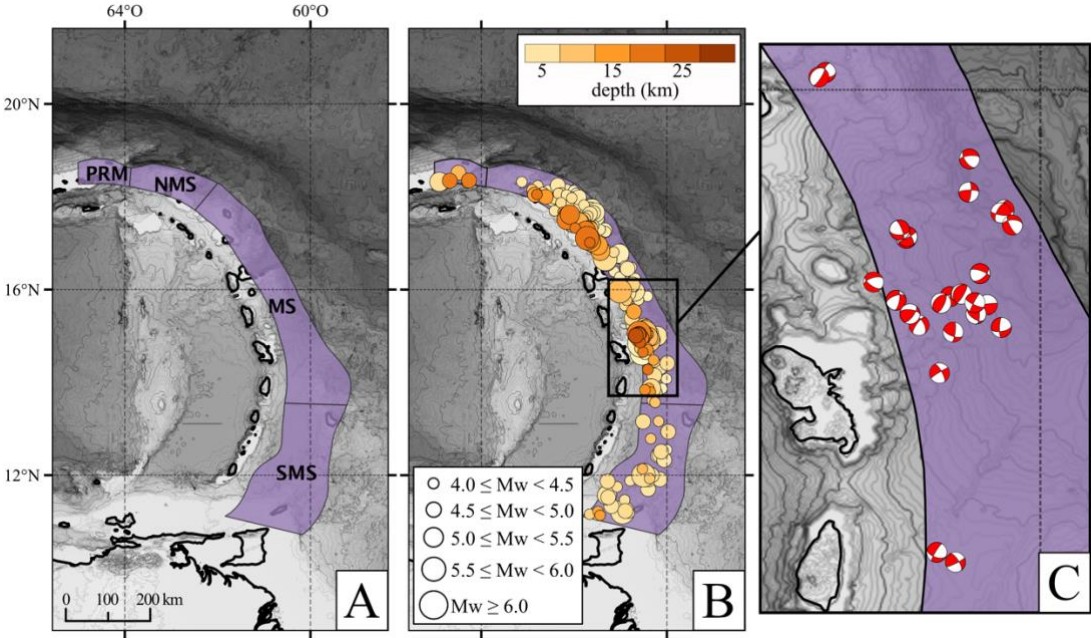

**Figure 10: Lesser Antilles mantle wedge seismotectonic zoning model. A, B and C: same as Fig. 5 but for the mantle wedge seismicity; MS: Mantle Source, NMS: North Mantle Source, PRM: Puerto Rico Mantle. Focal mechanisms from Ruiz et al. (2013).**

The seismogenic potential of the mantle wedge is poorly constrained due to the large earthquake location uncertainties and the lack of dedicated studies (in the Lesser Antilles and worldwide). An OBS study offshore Guadeloupe and Martinique recorded two M=3.1 and M=3.6 mantle wedge earthquakes, with different orientations and fault types, over a six-month period (Ruiz et al., 2013; **Fig. 10C**). The October 8, 1974, $M_S$=7.1–7.6 earthquake location is estimated in the arc lower crust by McCann et al., (1982) (**Fig. 2C**), along a NE-striking, SE-dipping normal fault, but its focal depth of 35 km suggests a possible mantle wedge event, as the Moho depth beneath Antigua is imaged at 30 km depth (Schlaphorst et al., 2021).Worldwide, the New-Zealand subduction shows the largest recorded mantle-wedge earthquake with a magnitude M=4.5 (Davey and Ristau, 2011). Current work is conducting to determine a proper seismic catalog based on waveform conversions (Foix et al., 2024).

**4.5 Volcanic seismotectonics and area sources**

Seismic hazard assessment for volcanic systems is challenging due to uncommon seismicity pattern laws, low earthquake magnitudes and high volcanic edifice heterogeneity affecting seismic wave attenuations and strong motion laws (e.g., Peruzza et al., 2017). Worldwide, volcano-tectonic earthquakes are generally limited to M<5–6 (McNutt and Roman, 2015), even though earthquakes up to M=6-7 have caused severe damages in Japan or Indonesia (Yokoyama, 2009) and have reached M=7.5–8 (e.g., 1990, Mount Pinatubo, Philippines).

The Lesser Antilles subduction system is marked by numerous volcanic eruptions and volcano-tectonic earthquakes. Seismo-volcanic crises were responsible for significant earthquakes and, in some cases, associated damages, such as the 1950-51 Nevis crisis ($M_W$=4.3) (Willmore, 1952), the 1966–67 Montserrat crisis with 32 felt earthquakes, the Guadeloupe Soufrière phreatic eruption in 1976-77 (M=4.6) or and unrest in 2017-2018 (ML=4.1) (Moretti et al., 2020). In order to account for these events, we define specific circular source areas associated with the active volcanic edifices and seismicity using a simple 10-km-radius definition and the crust thickness as depth limit (**Fig. 6**).

## 5 Discussion and recommendations

Recent studies proposed alternative methods to analyze seismotectonic. Mazzotti et al., (2011) incorporate geodetic data to derive strain rate and seismic moment rates. Beauval et al. (2018) use geologic and geodetic slip-rates to characterize a weakly instrumented zone. In this study, we propose an updated seismotectonic zoning model for seismic hazard assessment in the Lesser Antilles enriched by numerous recent improvements in the understanding of the regional seismotectonic context. We mixed traditional seismicity-based methods with geological and geodetic data. The zoning comprises area and fault sources for the upper plate and area sources for the subduction interface, the subducting plate, the mantle wedge, and volcanic centers. Major updates compare to previous studies (Martin and Combes, 2001; Bozzoni et al., 2011; Pagani et al., 2020b; Zimmerman et al., 2022) consist of: (*) better depth resolutions, owing to new slab and upper plate geometries; (*) a specific area source for the Marie-Galante graben, based on combined new tectonic, seismic, and geodetic data; (*) new propositions of mantle wedge and volcanic zoning; and (*) fully revised area sources for the subduction interface based on the integration of geodetic and coral data in a combined seismotectonic zoning model. Positioning area source boundaries may result in underestimating seismic hazard in short and long return periods, from 50 to $10^5$ years (Avital et al. 2018). To go further with PSHA calculations, we highly recommend adding an uncertainty on area source boundaries when it is possible. Alternative models may be used for PSHA estimations in regions with high levels of data and knowledge (e.g., California, Italia). In the Lesser Antilles each subset of data is incomplete, which does not allow us to propose alternative zoning.

As well as major updates listed above, we specifically compare our study to the seismotectonic zoning used as for the French Lesser Antilles PSHA (Martin and Combes, 2001), and propose entirely revised upper plate and intraslab zoning geometries (S6). New knowledge on the geological context, Moho and slab geometries, seismicity catalogues and rates, geodesy and seismic imaging allow us to propose a new zoning scheme. Only the trench and western volcanic arc zoning limits are consistent between the two models (**S6**). The greatest change lies in the fact that we divide our model according to the subduction structure (upper plate crust, interface, intraslab, mantle wedge, volcanoes) whereas Martin and Combes (2001) considered all the seismicity from 0 to 30 km depth to propose a "shallow" zoning. Recent seismotectonic zoning models

also consider this subduction structure division (Zimmerman et al, 2022) as seismicity of each domain presents specific characteristics, for instance in terms of maximum earthquake magnitude. Regarding outer rise maximum magnitude earthquake records around the world (e.g., Meng et al., 2012), we also decide to consider this region in our model, compared to Martin and Combes (2001). We also exclude the division of the back-arc region proposed by Martin and Combes (2001), as the sub-zones are not justified by a specific change in instrumental or historical seismic records or by specific fault data (activity, slip rate). Finally, our data set allows us to cover a larger area (**S6**). This zoning model can also be used in terms of seismic ground motion attenuation as wave attenuation laws (e.g., Youngs et al., 1997).

Our study also highlights major remaining epistemic uncertainties and unknowns that can impact seismic hazard assessment. Although the overall seismotectonics of the upper plate is relatively well defined in the few areas studied with dedicated surveys, given the very great structural heterogeneity observed. Specifics of most of the arc and forearc tectonics require further data collection and analyses. For example, an apparent velocity gradient of ~0.6 ± 0.3 and 1.4 ± 0.4 mm/yr for the eastern and northern components is observed in geodetic data between Saint-Lucia to Saint-Vincent-and-the-Grenadines, without an associated pulse of earthquake activity or known active faults. This points out the need for a more complete active fault and structural database of the whole Lesser Antilles arc and forearc.

Similarly, homogeneous earthquake catalogs with improved locations and magnitudes are an important need for both instrumental and historical seismicity. The ISCU-cat (Bertil, 2024) empirical magnitude conversion laws have been made from Ml or Md of the Trinidad seismological network (TRN) or from the Fort-de-France seismic station (FDF - IPGP network) to $M_W$ (Bertil et al., 2023). Conversions are calibrated on data recorded between 1986 and 2014. After 2014, local magnitude (Mlv) calculations from IPGP observatories differed (Massin et al., 2021) and TRN magnitude conversions have to be verified. Thus, Gutenberg-Richter distributions proposed in **S4** are more qualitative than quantitative. Changes in slope observed between magnitudes 3 and 4 could be a completeness effect, but also the result of inappropriate Md to $M_W$ conversions. In the Virgin Islands, no Md to $M_W$ conversion is available and Md = $M_W$ is directly used (Bertil et al., 2023). These Gutenberg-Richter distributions (**S4**) clearly illustrate that improvements in the magnitude conversions for M < 4.0 are still needed. We consequently describe this distribution in three levels of confidence: usable, questionable and unusable (**S3**).

One of the major sources of epistemic uncertainty concerns the subduction interface with its lack of instrumental large earthquake records and its overall very low geodetic interseismic coupling. However, we faced to a lack of geodetic measurements of this area which is mainly underwater, despite recent efforts. Worldwide, low interseismic coupling associated with limited large earthquake activity is observed in the Hellenic, Calabria, South Sandwich, and Mariana subduction zones (e.g., Vanneste and Larter, 2002; Vernant et al., 2014; Carafa et al., 2018). Yet, focal mechanisms indicate that the Lesser Antilles megathrust can generate moderate M=5–6 earthquakes, at least in its northern half. Coral data is sometime interpreted as indicative of deep (~40–65 km) high interseismic coupling (Philibosian et al., 2022) or interpreted as

coherent with geodetic vertical motions with observed subsidence (van Rijsingen et al., 2022). Issues of potential temporal variations of interseismic coupling, as in Mexico or Sumatra (Villafuerte et al., 2021; Philibosian et al., 2022), and the relationship between interseismic coupling and great earthquake potential (e.g., Kaneko et al., 2010) remain to be addressed. Efforts on seafloor geodesy measurements would help to enhance our understanding of the Lesser Antilles intercoupling behavior of the plate interface and we greatly recommend considering it for future oceanographic missions.

Finally, the uncommon seismicity sources identified in the mantle wedge and the volcanic centers also require dedicated studies before they can be fully integrated in seismic hazard assessments. In both cases, issues such as the earthquake maximum magnitudes and mechanisms, or appropriate ground motion attenuation laws demand dedicated global and, if possible, local studies. For the latter, the recognition that shallow moderate volcano-tectonic earthquakes can constitute a significant source of hazard implies further studies of the influence of eruptive phases on the triggering of volcano-tectonic earthquakes. The 10-km radius around each volcano should be revisited to be more specific to each volcano's characteristics. The creation of a specific volcano-tectonic earthquake catalog from instrumental and historical records is mandatory to better encompass their seismic activities. Volcanic seismicity from the two French volcanoes, La Soufrière and Mount-Pelée, can be downloaded from the IPGP observatory servers. The ISCU-cat may contain events from volcanic activity, such as the 1950 Mw=4.3 earthquake from the St. Kitts-Nevis seismo-volcanic crisis, or the 2020 Mw=3.6 earthquake from the St. Vincent seismo-volcanic crisis (Bertil pers. comm., Joseph et al., 2022). Seismic records may not be sufficient to calculate magnitude-frequency distributions and their associated attenuation laws.

**Supplement link**

**Team list**

Lesser Antilles Working Group: Marie-Paule Bouin[4], Éric Calais[5], Jean-Jacques Cornée[1], Nathalie Feuillet[4], Roser Hoste-Colomer[3], Mireille Laigle[6], Serge Lallemand[1], Jean-Frédéric Lebrun[1], Anne Lemoine[3], Boris Marcaillou[6], Mélody Philippon[1], Agathe Roullé[4], Claudio Satriano[4], Jean-Marie Saurel[4], Elenora van Rijsingen[7]

[4]Université de Paris, Institut de Physique du Globe de Paris, CNRS, F-75005 Paris, France.
[5]École normale supérieure, Département de Géosciences, CNRS, Paris, France.
[6]Géoazur, Université Côte d'Azur, CNRS, IRD, Observatoire de la Côte d'Azur.
[7]Utrecht University, Department of Earth Sciences, the Netherlands.

## Author contribution

**Océane Foix**: Conceptualization, Data curation, Formal analysis, Methodology, Software, Validation, Visualization, Original draft preparation, Review and editing. **Stéphane Mazzotti**: Conceptualization, Formal analysis, Funding acquisition, Methodology, Software, Visualization, Original draft preparation, Review. **Hervé Jomard**: Conceptualization, Formal analysis, Funding acquisition, Methodology, Original draft preparation, Review. **Didier Bertil**: Data curation. Formal analysis, Methodology, Original draft preparation, Review. **Lesser Antilles Working Group**: Data curation, Review.

## Competing interest

The authors declare that they have no known competing financial interests or personal relationships that could have appeared to influence the work reported in this paper.

## Acknowledgments

We express our gratitude to the Ministry of Ecological Transition and Territorial Cohesion for supporting this study (convention number 2201258550) across the ATLAS project, for the Lesser Antilles seismic hazard reassessment, and to the Epos-France consortium for its funding support. Lesser Antilles seismotectonic model detailed report and data are available on the OREME database (Foix et al., 2023b, https://data.oreme.org/alea/alea_data/antilles). E. Calais, from the Lesser Antilles Working Group, acknowledges support from the FEDER European Community program through the *Interreg Caraïbes 'PREST'* project (grant #5236), from the *Institut Universitaire de France*, and from the French National Research Agency (grant #ANR-21CE03-0010). The figures were produced using Generic Mapping Tool (Wessel et al., 2013), Affinity Designer, as well as Python and Microsoft Excel. We would like to thank the reviewers for their constructive criticism, which has helped to improve this paper.

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
