# Peer review of "Lesser Antilles Seismotectonic Zoning Model for Seismic Hazard Assessment"

_Natural Hazards and Earth System Sciences, 2024_

## Author Comment (AC1)

**Océane Foix**
UGA, ISTerre
1381 rue de la Piscine, 38610, Gières, France
oceane.foix@univ-grenoble-alpes.fr

**Editors of Natural Hazards
and Earth System Sciences**

**Subject**: Response to reviewers to the manuscript numbered NHESS-2024-53

August 2, 2024
Dear reviewer, editor,

Thank you for your feedback on our manuscript numbered NHESS-2024-53 and titled « Lesser Antilles Seismotectonic Zoning Model for Seismic Hazard Assessment ». Reviewer-1's comments have been addressed in point-by-point detailed answers below. Reviewer-1 comments are in **bold**, our replies are in normal text and changes we made in the manuscript are *green italic*.

We hope that you will find this revision favorable for publication in *Natural Hazards and Earth System Sciences*, and look forward to hearing from you.

Sincerely,
Océane Foix et al.

— — —

**Reviewer 1**

**(1) Foix et al. proposed an updated seismotectonic zoning model for the Lesser Antilles. This study benefits from a variety of recent findings by others on such as seismicity, focal mechanisms, and geodetic observations. While summarizing these findings and using them as a basis to define seismotectonic zones is beneficial, the article itself does not clearly illustrate how these data are used, or how the zoning is defined quantitatively.**

We understand that the data used and the zoning description resume provided in the manuscript may seem insufficient for the reader. However, all these descriptions are detailed in the French Ministry report (Foix et al., 2023), cited in the article. Our aim was to propose something lighter for the reader, as all details are already given in the report, to capture the essence of our ideas. To provide context and as a reminder, this work was funded by the Risk Prevention Department of the French Ministry (Ministry of Ecological Transition and Territorial Cohesion). As a result of this funding, a report was mandatory (62 pages) and it does not benefit from peer reviews. To better illustrate how we used the data, we proposed the following modifications in the method section:

**l.127**: *Seismicity distribution, deformation style extracted from focal mechanisms, crustal fault locations and tectonic features were superimposed using Geographic Information Systems (GIS) tool as illustrated in Fig. 4, 7, 8 and 9. Boundaries depict a consensus between seismotectonic data and discussions with Lesser Antilles experts (Lesser Antilles Working Group).*

Seismotectonic zonings are classically built on the crossing of more or less quantitative criteria, with subjective interpretation. In areas where knowledge is sufficient (data collected homogeneously over space and time), we can move towards purely quantitative criteria, or even do without zoning by considering smoothed seismicity, for example. Zoning is a tool for managing lack of knowledge and the Lesser Antilles arc is generally poorly known. The most recent data have been collected and integrated for this study. To define quantitative variations such as the Gutenberg-Richter parameters and the deformation rate for each area, we need a proper seismic catalog. The catalog we used still needs some improvement and this work is actually in progress. However, we're a long way from being able to propose something purely quantitative. We agree that we did not discuss that in the manuscript and we propose to add the following lines:

**l.118**: *Seismotectonic zonings are designed to fill the gaps in our knowledge, and the Lesser Antilles arc and its active seismicity is poorly known. Area sources are built by crossing more or less quantitative criteria with subjective interpretation. When knowledge is sufficient (data collected homogeneously over space and time) the zoning can be purely based on quantitative criteria. In this study, the most recent data has been collected and integrated.*

**l.140**: *The ISCU-cat spatial seismicity rate variations are analyzed to determine specific activity changes and then propose area source boundaries. Gutenberg-Richter analysis highlights slope variations that can be induced by magnitude conversion used for magnitude homogenization. The ISCU-cat needs for magnitude estimation improvements are discussed in the discussion (section **4**).*

**l.480**: *The ISCU-cat (used in this study) empirical magnitude conversion laws have been made from Ml or Md of the Trinidad seismological network (TRN) or from the Fort-de-France seismic station (FDF - IPGP network) to Mw (Bertil et al., 2023). Conversions are calibrated on data recorded between 1986 and 2014. After 2014, local magnitude (Mlv) calculations from IPGP observatories differed (Massin et al., 2021) and TRN magnitude conversions have to be verified. Thus, Gutenberg-Richter distributions proposed in **S3** are more qualitative rather than quantitative. Changes in slope observed between magnitudes 3 and 4 could be a completeness effect, but also the result of inappropriate Md to Mw conversions. In the Virgin Islands, no Md to Mw conversion is available and Md = Mw is directly used (Bertil et al., 2023). Gutenberg-Richter distributions (S3) well illustrate that improvements in the magnitude conversions for M < 4.0 are still needed.*

**(2)** **Section 3.1 states that three principles are followed, but it is not clear in the data and methods section how seismicity distribution, for instance, is used for zoning in different**

**parts of the subduction zone. Is it based on depth, or spatial clustering? These aspects need to be introduced in the methods section.**

Instrumental seismicity hypocenters were analyzed to distinguish relative increase or decrease in the seismicity rate distribution in the different subduction zone parts (arc, fore-arc, basin, accretionary wedge, subducting interface and intraslab regions) to help us to define zoning boundaries. The instrumental seismic catalog state of progress does not allow us to go deep into the analysis (solid Gutenberg Richter calculations, deformation rate estimations). From historical seismicity, studies are still in progress to better optimize event interpreted depth locations. However, earthquake geographical possible locations help us to discuss past activity of the region without being able to solve if the event was part of the crust, interface or intraslab region. We agree that these aspects need to be introduced in the method section for more clarity. We proposed the following modifications:

**l.140**: Same as comment (1)

**l.153**: *Historical seismicity allows us to discuss past activity of specific regions and to compare it with instrumental seismicity and geodetic data (Foix et al., 2023).*

**(3) A new seismotectonic zoning model would be useful for seismic hazard assessment. In the current manuscript, it is not clear whether there was a previous zoning model for the Lesser Antilles. If so, a comparison with the previous version would help to identify the contributions from the recent findings. It would be even better if a preliminary hazard model could be provided and compared with the 2002 version, as this would enrich the discussion.**

The hazard model calculation would be the work of the next 2 years by another team. A preliminary comparison for the Guadeloupe area was considered but not adopted for various reasons. In particular, the zoning includes new sources that require considerable work to produce an interesting result. The ground motion prediction laws are no longer valid, the seismic catalog cannot be used right now and an appropriate magnitude conversion scale is needed. The new catalog that will serve as the basis for the calculation has not been completed yet. Indeed, we tried to complete the ISCU-cat for M ≤ 3 with the IPGP observatory seismic catalogs for the Guadeloupe area, but magnitude conversion laws are inappropriate (**figure A**).

Instead, we chose to compare seismotectonic models. We clarified in the manuscript that "*Seismic hazard models are few (e.g., Bozzoni et al., 2011), generally at the scale of the whole Caribbean region with a focus on the Greater Antilles (Pagani et al., 2020b; Zimmerman et al., 2022). Previous probabilistic seismic hazard assessments were conducted in 2002 for the Lesser Antilles (Martin and Combes, 2001).*" at **l.34**. We agree that a specific comparison of our study with the previous zoning would help to capture improvement and changes. In this sense, we add comments in the discussion section and a figure in the annex with model comparison.

[Figure]

**Figure A**: ISCU-cat (blue) and IPGP (red) md (left) and Mw (right) comparison.

**Annex S6 Martin and Combes (2001) vs this study seismotectonic models**: *Comparison between the Lesser Antilles seismotectonic model proposed in this study (green) and the previous model requested by the Ministry for the seismic hazard assessment and produced by Martin and Combes (2001 - yellow dashed lines). We compare models for the upper plate crust (this study) and superficial depths (< 30 km depths) in A, and for the intraslab in B.*

[Figure]

**l.37**: *The authors' seismotectonic model and resulting PSHA calculation were, as this study, a ministerial request. Their superficial zoning was based on gravimetric, magnetic, geologic,*

*seismic and topo-bathymetric data from 0 to 30 km depth to divide the area into homogeneous domains. The subduction zoning was based on plate interface dip variations induced by the presence of ridges and fractures (Martin and Combes, 2001). No specific zoning for the plate interface was proposed.*

**l.456**: *As well as major updates listed above, we specifically compare our study to the seismotectonic zoning used as reference for the French Lesser Antilles PSHA (Martin and Combes, 2001), and proposed an entirely revised upper plate and intraslab zoning geometries (**S6**). New knowledge on geological, Moho and slab geometries, seismicity rate records, geodesy and seismic imaging induced changes in zoning limits. Only the trench and west volcanic arc end structures used as zoning limits are consistent between the two models (**S6**). The greatest change lies in the fact that we divided our model according to subducting structure (upper plate crust, interface, intraslab, mantle wedge, volcanoes) whereas Martin and Combes, (2001) mixed their seismicity from 0 to 30 km depth to propose a "shallow" zoning. Recent seismotectonic models are considering this subduction structure division (Zimmerman et al, 2022) as seismicity of each region presents specific characteristics as wave attenuation laws (e.g., Youngs et al., 1997). Regarding outer rise maximum magnitude earthquake records around the world (e.g., Meng et al., 2012), we also decide to consider this region in our model, compared to Martin and Combes, (2001). Unfortunately, the poor instrumental seismicity record will not allow us to determine deformation rate, useful for PSHA estimations (**S2** and **S3**). We exclude the division of the back-arc region as proposed by Martin and Combes, (2001) as no specific change in instrumental or historical seismic records is highlighted (**Fig. 5**), and as no faults with sustainable activity and estimated velocity rate allow us to propose smaller area sources. Finally, our data set allows us to cover a larger area (**S6**).*

**(4) The modeling work to explore interseismic coupling on the plate interface is interesting and could be included in the main text. The observation and modeling support the claim that more data covering the plate interface, potentially from seafloor geodesy, are needed to resolve the coupling issue.**

We agree that the interseismic coupling exploration is interesting and could be added to the main text, and at the same time, we think that this study does not go deep enough to bring it to the forefront. More explorations are needed with 3D modeling and more variations in the coupling value and in the coupling depth. This work is beyond the scope of this study. At the same time, we agree that it is important that new data from seafloor geodesy are obtained and we have added this information into the manuscript.

**l.500**: *Efforts on seafloor geodesy measurements would help to enhance our understanding of the Lesser Antilles intercoupling behavior of the plate interface and we greatly recommend considering it for future oceanographic missions.*

**(5) The current writing is more like a report than an article. Line 590 and 595 mention a report and dataset that were already published by the authors. How does the current submitted article differ from that report?**

We agree with the comment and tried to modify the text to fit more with an "article style" than a report. We tried to emphasize the problems around the mantle wedge and volcanic zoning, and around the Marie-Galante graben in the introduction. A previous report exists which was mandated by the French ministry and that does not include peer reviews. The Marie-Galante graben analysis is new and not in the report.

**l.46**: *Previous seismotectonic models of the Lesser Antilles did not consider mantle wedge seismicity and did not integrate specific zoning for the volcanic seismicity (Martin and Combes, 2001; Pagani et al., 2020b; Zimmerman et al., 2022). Due to its physical properties, the mantle wedge is not considered as a site for seismic nucleation. Seismicity is generally weak and diffuse (Hasegawa et al. 2009) but recent works indicate more sustained activity in (New Zealand, Davey and Ristau (2011); Japan, Uchida et al. (2010); Alps, Malusa et al. (2016); Lesser Antilles, Laigle et al., (2013)). The $M_w$=4.5 in New-Zealand (Davey and Ristau, 2011) and a possible 1974 M=6.9-7.5 event in the Lesser Antilles (McCann et al., 1982) raise the question of the importance of considering this seismicity for seismic hazard assessment. PSHA allocates this seismicity to the other seismogenic sources, i.e. the crusts of the upper or lower plate, or along the interface. This means that the distances between the hypocenters and the hazard calculation sites are poorly resolved. Moreover, working on PSHA at volcanic regions is a challenge regarding earthquake characteristics: low magnitude and high seismic wave attenuation. In the Lesser Antilles, the Nevis crisis of 1950-51 caused damage to buildings, with a maximum magnitude of Mw = 4.3 (ISC catalog) and intensity VIII (Willmore 1952). It is therefore important to be able to propose a way to consider it.*

**l.63**: *We propose a specific focus on the Marie-Galante graben where we estimate and compare the extensional rate from seismic and geodetic data.*

**(6) Line 120-125, did you include focal mechanisms from Lindner et al., 2022?**
We did not specify the time range of our FMAnt21 catalog and we agree that this could be confusing. The FMAnt21 catalog starts in 1977 and ends in 2021. We have added this information at **l.157**. At the time we did our analysis to compute average faulting types, we were not aware of Lindner et al. (2023) study. However, we had the chance to exchange information with him, and we are planning to update the FMAnt-21 catalog using more recent data and new articles as the really useful one of Lindner et al. (2023). Updating the catalog will be one of our priorities for the next few months.

**l.155**: *We construct a composite catalog of earthquake focal mechanisms comprising 572 events from the GMCT (Dziewonski et al., 1981; Ekström et al., 2012), ISC (Letas, 2018; Letas et al., 2019) as well as Corbeau et al. (2019, 2021), González et al. (2017), and Ruiz et al. (2013), **from 1977 to 2021**, hereafter named as FMAnt2021 (Focal Mechanisms Antilles 2021).*

**(7) Line 130, plate interface geometry plays an important role in your work. How are the two slab models unified? This needs to be detailed in the main text or the appendix.**

We agree that the description of the slab top geometry is too short and needs more details in the text and a figure in the appendix.

**l.162**: *We have georeferenced and digitized these interface geometries to combine them in one unique surface using a GIS tool every 10 km of depth. We then transformed it into a grid that can be extrapolated to get a surface and to be used for earthquake sorting. Paulatto et al. (2017) slab and Moho geometries were not used in order to keep consistency along the arc.*

**S1 Slab top geometry unification from Laurencin et al. (2018) and Bie et al. (2020) in the Lesser Antilles** : **A**: Slab top geometries from Laurencin et al. (2018 – pink dotted lines), Bie et al. (2020 – purple dotted lines) and of the unified slab (black solid lines). **B** and **C** are zooms of the unified region with Laurencin et al. (2018) and Bie et al. (2020) geometries in **B** and the unified slab geometry in **C**.

[Figure]

**(8) Line 365, Bie et al., 2022 presented a seismic velocity model that supports the cold mantle wedge nose.**

Thanks. We have added the citation and also included the Halpaap et al. (2019) in the mantle wedge section which describes mantle wedge seismicity around the world and proposes the escape of fluids from the plate interface and slab to the mantle wedge.

**l.409**: *The Lesser Antilles subduction is characterized by a cold mantle wedge associated with normal-faulting seismicity (Bie et al., 2020, **2022**; Laigle et al., 2013b; Ruiz et al., 2013), as also observed in the Greek, New Zealand, and Northern Japan subductions (Uchida et al., 2010; Davey and Ristau, 2011; **Halpaap et al., 2019, 2021**).*

**l.411**: *This peculiar "supra-slab" seismicity may be explained by the presence of pyroxenitic material within peridotites instead of aseismic serpentinized peridotite (Laigle et al., 2013b), or by subduction fluids from slab source expelled to the mantle wedge (Haalpap et al., 2019) and which could result in a cold mantle wedge (Hicks et al., 2023).*

**(9) Line 395, a circular source area with a 10-km radius is defined for all active volcanoes. What is the logic behind selecting 10 km? Is it arbitrary? Should some physical properties be considered in deciding this number? This part warrants a discussion in the final section.**

In this study, we provide general information that should be considered for the Lesser Antilles volcano-related seismic hazard assessment as it was not considered before. We propose a first step that aims to define specific zones associated with the volcanic edifices and their seismic activity. In that sense, we propose a simple definition consisting of a 10 km radius circle around each edifice and the crust thickness as depth limit, in order to include all potential seismicity related to volcanic activity. This decision was made in agreement with researchers working at the Guadeloupe and Martinique volcanic observatories (e.i., J.M. Saurel) regarding the seismicity locations of the past volcano-tectonic events. We verify each volcano to ensure that all of them (their surface edifice footprint) are contained within this radius. However, a deep analysis of the volcano-tectonic earthquake distributions will be essential to prevent any exclusion of events outside the 10 km radius. We agree that this part needs a section in the discussion to clarify epistemic uncertainties and identify which work needs to be done. We have already specified **l.502** "*Finally, the uncommon seismicity sources identified in the mantle wedge and the volcanic centers also require dedicated studies before they can be fully integrated in seismic hazard assessments. In both cases, issues such as the earthquake maximum magnitudes and mechanisms, or appropriate ground motion attenuation laws demand dedicated global and, if possible, local studies. For the latter, the recognition that shallow moderate volcano-tectonic earthquakes can constitute a significant source of hazard implies further studies of the influence of eruptive phases on the triggering of volcano-tectonic earthquakes*" and we have added:

**l.509**: *The 10-km radius around each volcano should be revisited to be more specific to each volcano's characteristics. The creation of a specific volcano-tectonic earthquake catalog from instrumental and historical records is mandatory to better encompass their seismic activities. Volcanic seismicity from the two French volcanoes, La Soufrière and Mount-Pelée, can be downloaded from the IPGP observatory servers. The ISCU-cat may contain events from volcanic activity, such as the 1950 Mw=4.3 earthquake from the St. Kitts-Nevis seismo-volcanic crisis, or the 2020 Mw=3.6 earthquake from the St. Vincent seismo-volcanic crisis (Bertil pers. comm., Joseph et al., 2022). Seismic records may not be sufficient to calculate magnitude-frequency distributions and their associated attenuation laws.*

**(10) The authors may consider depositing their homogenized seismic catalog for open access.**

The ISCU-cat is a Mw unified catalog for Mw ≥ 3.0, mainly extracted from the ISC database (Bertil et al., 2023). The aim of this catalog is to better illustrate and understand the spatial distribution of the Caribbean arc seismicity, in a more homogeneous way than other regional catalogs. Indeed, the catalogs from IPGP observatories (Martinique and Guadeloupe islands) are incomplete north of the Virgin Islands and south Saint Lucia, and have not been shared with the ISC since 2015. Seismic catalogs from other regional organizations only cover part of the arc. The other goal of this ISCU-cat is to analyze the strongest magnitudes, observe how the earthquake detection threshold evolves over time and make comparisons with the seismicity of the Sisfrance Antilles catalog (Bertil et al, 2023). This catalog needs to be completed for Mw < 3.0, with relocations from the CDSA 1971-2013 catalog (Massin et al, 2021) and with IPGP observatories data after 2014.

The ISCU-cat is not a product of this article and we apologize if it was understood that way. A specific citation was provided **l.133** : Bertil et al., (2023), which is related to a poster. We have modified the sentence ("and built by Bertil et al., (2023)") and we have added one more citation at **l.137** to prevent any confusion. This catalog will probably be shared at the end of the year with a proper DOI on the BRGM (Bureau de Recherches Géologiques et Minières) data server. The catalog that will be used for the future PSHA is not the ISCU-cat but will be the work of Gonzalez, Corbeau, Satriano and IPGP observatories, and should be available in 2025.

— — —

In addition to Reviewer 1 comments, we also correct:
- Hough (2013) instead of 2023 (**l.329**)
- M instead of Mw for the 1839 and 1843 events (**l.33, 270, 327**)
- 2018-09-28 12:32 instead of 13:32 in the FMant-2021 catalog
- Geoter (2002) reference to Martin and Combes (2001)
- and add discussions on data uncertainties :
  - **l.149**: *The exact location of some historical events is still debated, such as the 1839 and 1843 earthquakes. Bernard & Lambert (1988) and McCann et al. (1984) interpreted them as megathrust earthquakes whereas van Rijsingen et al. (2021) proposed that the 1843 event had a smaller magnitude, or different mechanism or location within the subducted slab, and that the 1839 event could also be located in the subducted slab.*
  - **l.165**: *The slab top geometry may vary according to the publications, but also according to the interpretation of the presence or absence of a slab. Beneath the Lesser Antilles central area, at depths of around 170-200 km, Lindner et al. (2023) observed a seismic gap in the lithosphere of subducted American plates, whereas Braszus et al. (2021) observed a continuous slab, in agreement with the tomography of Bie et al. (2020). These differences would have an impact on earthquake sorting and the resulting statistics.*
  - **l.174**: *Various unknowns and interpretations remain on fault activities. The Anegada passage fault system (**Fig. 3 (5)**) motion was interpreted from extensional faulting to sinistral or dextral transtension (Laurencin et al., 2017*

*and references therein). Fault activity is sometimes debated, such as for the V-shaped basin faults from Guadeloupe to Saint-Kitts-and-Nevis (Feuillet et al., 2001, 2011a ; Boucard et al., 2021). Moreover, structures still need to be imaged and understood south of Saint-Lucia.*

---

## Author Comment (AC3)

**Océane Foix**
UGA, ISTerre
1381 rue de la Piscine, 38610, Gières, France
oceane.foix@univ-grenoble-alpes.fr

**Editors of Natural Hazards
and Earth System Sciences**

**Subject**: Response to reviewer RC2 to the manuscript numbered NHESS-2024-53

November 28, 2024
Dear reviewer RC2, editor,

Thank you for your feedback on our manuscript numbered NHESS-2024-53 and titled « Lesser Antilles Seismotectonic Zoning Model for Seismic Hazard Assessment ». Reviewer-2's comments have been addressed in point-by-point detailed answers below. Reviewer-2 comments are in **bold**, our replies are in normal text and changes we made in the manuscript are *green italic*.

We hope that you will find this revision favorable for publication in *Natural Hazards and Earth System Sciences*, and look forward to hearing from you.

Sincerely,
Océane Foix et al.

— — —

**Reviewer 2**

**(1) It took me a while to realize that the manuscript is a condensed report that was previously submitted to a funding source (Foix et al., 2023a). It would be helpful to the reader to clearly define the assignment and parameters of this report.**

We agree that the link between Foix et al., (2023) and this study are not clear and need to be precise. Foix et al., (2023), cited in this article, present most of these descriptions in detail. Our aim was to propose something lighter for the reader to capture the essence of our ideas and provide focuses on specific research aspects relevant to Lesser Antilles seismic hazard (e.g., Marie-Galante graben analysis). This work was funded by the Risk Prevention Department of the French Ministry (Ministry of Ecological Transition and Territorial Cohesion). As a result of this funding, a report was mandatory (62 pages, Foix et al., 2023) and it does not benefit from peer reviews. We add the following lines in the beginning of the manuscript to clarify these points:

*l.63 In this study, we present a new seismotectonic zoning model built for Lesser Antilles, with seismogenic source characteristics provided for future seismic hazard assessment. The seismotectonic zoning model comprise the Lesser Antilles upper plate, subducting oceanic plate, subduction interface, mantle wedge, and volcanoes, based on a compilation and*

*reanalysis of seismicity and fault catalogs, earthquake focal mechanisms, and geodetic data. This work was carried out in response to a request from government authorities and is the subject of a detailed technical report (Foix et al., 2023). Here, we present the main scientific points leading to the zoning model, as well as a focus on the Marie-Galante graben where we estimate and compare the extension rates from seismic and geodetic data.*

**(2) The manuscript promises to be an improvement over past assessments in the area, although nowhere it is specified what the past assessments were and whether their aims and methodologies were the same. The only past assessments listed are Pagani (global assessment), Zimmerman (Caribbean-wide) and Geoter (France, but hard to tell because of the partial reference).**

Pagani, Zimmerman and Geoter are the only existing past assessments. The Geoter work is the only study that specifically treats the Lesser Antilles arc and not the whole Caribbean region. We agree that the Geoter referee is only a partial reference and we replace it with Martin and Combes (2001) into the manuscript. In reply to RC1's comments, we also add comparison and methodology details between our study and Martin and Combes (2001) :

**l.44**: *Previous PSHA of the French islands was conducted in 2001 (Martin and Combes, 2001). This seismotectonic zoning model and resulting PSHA calculation were, as this study, a request from governmental authorities to serve as a basis for the national seismic zoning revision process. The crustal zoning was primarily based on structural data (gravimetric, magnetic, geologic, seismic and topo-bathymetric data from 0 to 30 km depth). The subducting plate zoning was based on plate interface dip variations induced by the presence of ridges and fractures. No specific zoning for the plate interface was proposed.*

**l.488**: *As well as major updates listed above, we specifically compare our study to the seismotectonic zoning used as for the French Lesser Antilles PSHA (Martin and Combes, 2001), and propose entirely revised upper plate and intraslab zoning geometries (S6). New knowledge on the geological context, Moho and slab geometries, seismicity catalogues and rates, geodesy and seismic imaging allow us to propose a new zoning scheme. Only the trench and western volcanic arc zoning limits are consistent between the two models (S6). The greatest change lies in the fact that we divide our model according to the subduction structure (upper plate crust, interface, intraslab, mantle wedge, volcanoes) whereas Martin and Combes (2001) considered all the seismicity from 0 to 30 km depth to propose a "shallow" zoning. Recent seismotectonic zoning models also consider this subduction structure division (Zimmerman et al, 2022) as seismicity of each domain presents specific characteristics, for instance in terms of maximum earthquake magnitude. Regarding outer rise maximum magnitude earthquake records around the world (e.g., Meng et al., 2012), we also decide to consider this region in our model, compared to Martin and Combes (2001). We also exclude the division of the back-arc region proposed by Martin and Combes (2001), as the sub-zones are not justified by a specific change in instrumental or historical seismic records or by specific fault data (activity, slip rate). Finally, our data set allows us to cover a larger area (S6). This*

*zoning model can also be used in terms of seismic ground motion attenuation as wave attenuation laws (e.g., Youngs et al., 1997).*

**Annex S6 Martin and Combes (2001) vs. this study seismotectonic zoning models**: *Comparison between the Lesser Antilles seismotectonic zoning model proposed in this study (green) and the previous model produced by Martin and Combes (2001 - yellow dashed lines), both requested by the Ministry for the seismic hazard assessment. We compare models for the upper plate crust (this study) and superficial depths (< 30 km depths) in A, and for the intraslab in B.*

[Figure]

**(3)** **If I am not mistaken, the focus of the paper is on defining a seismotectonic zoning model for the Lesser Antilles (although the abstract says "seismotectonic model and zoning" (L16). How is "model" defined? Is it simply the authors' interpreting the geophysical evidence as belonging to different tectonic regimes?**

We agree that "seismotectonic model and zoning" could be confusing. In this article, we present a new zoning, based on a seismotectonic model derived from a compilation and interpretation of existing data detailed in Foix et al. (2023), with specific points discussed here. In seismic hazard studies and projects, "seismotectonic zoning models" are classically built on the crossing of more or less quantitative criteria with subjective interpretation, which is not that simple. In this context, "model" is viewed in its most general meaning (postulates, data, and inferences used to describe an entire system) rather than its numerical sense. In theory, in areas where knowledge is sufficient (data collected homogeneously over space and time), we can move towards purely quantitative criteria, or even do without zoning by considering smoothed seismicity, for example. In practice, zoning is a tool for managing the lack of knowledge. And the Lesser Antilles arc is generally poorly known. The most recent data have been collected and integrated for this study.

**(4) Most of the work appears to be compilations of catalogs and of past studies. The only original work includes the magnitude frequency distributions, the calculations of b-values and the 2-D geodetic modeling, all of which are inexplicably hidden in the Supplement (S2-S4).**

Our study is indeed based on a compilation of catalogs and past studies, which allow us to propose a totally new and revisited seismotectonic zoning model, a fundamental basis for seismic hazard assessment of the Lesser Antilles (e.g., mandatory for the future Eurocode-8). This new zoning model is the main result and it needs to be communicated to the seismic hazard community. The magnitude frequency distribution (MFD) is also a building block for PSHA calculation, but, as it is, the present seismic catalog cannot be directly used and ongoing work is done on a final catalog. Our work on MFD helps highlight the limits of the current catalog and requirements for a new one. Thus, our MDF (and consequently b-value calculation) needs improvement and we decide to only feature it in supplements. As for RC1's comments, we add in the text:

**l.147**: *The ISCU-cat spatial variations in seismicity are analyzed to determine area source boundaries and their associated Gutenberg-Richter distributions. The ISCU-cat needs for magnitude estimation improvements are discussed in section **5**.*

**l.512**: *The ISCU-cat (Bertil, 2024) empirical magnitude conversion laws have been made from Ml or Md of the Trinidad seismological network (TRN) or from the Fort-de-France seismic station (FDF - IPGP network) to Mw (Bertil et al., 2023). Conversions are calibrated on data recorded between 1986 and 2014. After 2014, local magnitude (Mlv) calculations from IPGP observatories differed (Massin et al., 2021) and TRN magnitude conversions have to be verified. Thus, Gutenberg-Richter distributions proposed in **S4** are more qualitative than quantitative. Changes in slope observed between magnitudes 3 and 4 could be a completeness effect, but also the result of inappropriate Md to Mw conversions. In the Virgin Islands, no Md to Mw conversion is available and Md = Mw is directly used (Bertil et al., 2023). These Gutenberg-Richter distributions (**S4**) clearly illustrate that improvements in the magnitude conversions for M < 4.0 are still needed. We consequently describe this distribution in three levels of confidence: usable, questionable and unusable (**S3**).*

We agree that the interseismic coupling exploration is interesting and could be added to the main text, and at the same time, more explorations are needed with 3D modeling and variations in the coupling value and coupling depth. In that sense, we add the figure into the main text, but we did not spend more space in detailing this work.

However, the Marie-Galante graben analysis is new and uses geodetic as well as MFD to determine extension rate and compare them (section **4.2.4**). We also provide the first zoning for the mantle wedge (compared to all other seismotectonic zoning for subduction around the world), and propose a zoning for the active volcanoes (for the Lesser Antilles). We agree this was not clear and highlighted in the manuscript and, as for RC1's comments, we add the following lines:

**l.50**: *Previous seismotectonic zoning models of the Lesser Antilles did not consider mantle wedge seismicity and did not integrate specific zoning for the volcanic seismicity (Martin and Combes, 2001; Pagani et al., 2020b; Zimmerman et al., 2022). Due to its physical properties, the mantle wedge is generally not considered as a site for seismic nucleation, with only weak and diffuse seismicity (Hasegawa et al. 2009). However, recent works indicate more sustained activity (New Zealand, Davey and Ristau (2011); Japan, Uchida et al. (2010); Alps, Malusa et al. (2016); Lesser Antilles, Laigle et al., (2013)). The $M_w$ = 4.5 in New-Zealand (Davey and Ristau, 2011) and a possible 1974 M = 6.9-7.5 event in the Lesser Antilles (McCann et al., 1982) raise the question of the importance of considering this seismicity for seismic hazard assessment. Allocating this seismicity to the other seismogenic sources (i.e. upper plate crust, lower plate, or the subduction interface) results in biased hypocenter distances hazard calculation. Moreover, working on PSHA in volcanic regions is a challenge regarding earthquake characteristics: low magnitude and high seismic wave attenuation. In the Lesser Antilles, the Nevis crisis of 1950-51 caused damage to buildings, with a maximum magnitude of Mw = 4.3 (ISC catalog) and intensity VIII (Willmore 1952). It is therefore important to be able to propose a way to consider it.*

**(5) Given the sparse data on which the zonation is based, their interpretation is likely not definitive, yet there is little discussion about alternatives and implications of the proposed zonation.**

Indeed, seismotectonic zoning models need to be continuously revisited in the light of new knowledge and data to propose up to date versions (section **5** - discussion). These revisions are classically motivated by new rules and regulations. In our case, the new Eurocode-8 is motivating the revision of the model (Foix et al., 2023). We agree that, at this stage, we do not discuss the implications of our new seismotectonic zoning model for PSHA calculations (this work will start in early 2025). Concretely, our model and its area source boundaries will have impacts on the MFD (b-value, seismicity rates). Unfortunately, evaluating this impact with the current instrumental catalog is useless as the ISCU-cat will not be the one used for the PSHA calculations. Furthermore, for the seismic hazard estimations, we highly recommend adding an uncertainty on area source boundaries not related to a specific structure (subduction trench, accretionary wedge limits etc), with specific weights on the parameters used for the PSHA calculation. To propose and evaluate alternative models, a possibility could be to design one model for each parameter (seismicity, geodesy, tectonic etc.) and compare and test their effects on PSHA calculation. Moreover, we encourage a comparison between this input model and previous ones, even if it is beyond the scope of this study. We add the following lines to the discussion:

**l.471**: *Recent studies proposed alternative methods to analyze seismotectonic. Mazzotti et al., (2011) incorporate geodetic data to derive strain rate and seismic moment rates. Beauval et*

*al. (2018) use geologic and geodetic slip-rates to characterize a weakly instrumented zone. In this study, we propose an updated seismotectonic zoning model for seismic hazard assessment in the Lesser Antilles enriched by numerous recent improvements in the understanding of the regional seismotectonic context. We mixed traditional seismicity-based methods with geological and geodetic data.*

**l.481**: *Positioning area source boundaries may result in underestimating seismic hazard in short and long return periods, from 50 to $10^5$ years (Avital et al. 2018). To go further with PSHA calculations, we highly recommend adding an uncertainty on area source boundaries when it is possible. Alternative models may be used for PSHA estimations in regions with high levels of data and knowledge (e.g., California, Italia). In the Lesser Antilles each subset of data is incomplete, which does not allow us to propose alternative zoning.*

**(6) A more fundamental question is whether it is useful to divide the study area into so many zones given the sparse instrumental and historical seismicity, and the geodetic data available for analysis. Wouldn't it be simpler to divide the north and south parts of the Lesser Antilles arc into Outer Rise, intraslab, slab interface, sub-crustal and crustal zones? This is particularly true because the zonation exercise is meant as a first step for hazard, ground shaking, and damage assessments, each of which will likely have large uncertainties. The effort should therefore be to minimize the propagation of uncertainties even at the expense of lumping together several "zones" to achieve a more robust characterization of each "mega-zone".**

We understand your suggestion to simplify the seismotectonic model by grouping the area sources together, as Zimmerman et al. (2022) or Bozzoni et al. (2011), to limit the uncertainties associated with the data scarcity. However, such an option would erase fundamental differences in tectonic styles and rates between different regions, clearly illustrated by the variations in focal mechanisms, seismicity level, known structures, etc., which are the basis of a tectonic model. We have opted for a finer division of the Lesser Antilles to take into account the geological and tectonic complexity of the region with variation in deformation regimes (Foix et al., 2023 and figure 9c of this study) or fault distributions (figure 3). Oversimplification of these specificities will have a direct impact on PSHA estimation. Indeed, zoning aims to capture spatial variations in order to best reflect local particularities, essential for hazard and damage assessments. This allows us to assign zone-specific seismicity rates and maximum magnitude. Future PSHA studies would have the opportunity to compare hazard estimates provided by a fine division (this study) or a north-south division like Bozzoni et al. (2011). Moreover, data scarcity does indeed lead to uncertainty. The division into several zones enables us to better constrain each source and to manage uncertainty more rigorously, by limiting the propagation of errors at the global level. Grouping area sources may reduce the reliability of results in local contexts. This methodology is also motivated by the need to compare our model with other similar studies carried out in complex tectonic contexts, here the previous

seismotectonic zoning model conducted by Martin and Combes (2001). The method we use allows us to provide a model adapted to the specificities of the Lesser Antilles, often neglected in global models. To handle this questioning, we add a short comment in the methodology part:

**l.134**: *To address geological and tectonic complexities and specificities of the region, we have opted for a division into several zones instead of a simple north-south division. Specific detailed subdivisions are limited to cases where they are supported by the available data. This methodology better constraints each source and allows us to manage uncertainties more precisely, by limiting the propagation of errors to the whole region.*

**(7) To turn the manuscript into a paper suitable for global readership, I would recommend:**

- **Clarifying what past assessments were and what your improvements are**: Done (report to point 2)
- **Moving the Methods and data into its own heading before the Seismotectonic zoning model section**: Done
- **Clearly stating what you mean by "seismotectonic zoning model" or simply call it "proposed seismotectonic zonation for the Lesser Antilles":** Done - seismotectonic zoning model refer to earthquake source characterization (refer to the last point of this section)
- **Including the original work, shown in Supplement S2-S4 in the main part of the paper.** We include S4 as explained in point 4.
- **The discussion chapter is weak. Please strengthen it with a discussion of alternative zonations and their implications to seismic hazard**: with the help of RC1' and RC2's comments we greatly improve our discussion chapter, adding comparison between previous and our study models, a discussion on the used seismic catalog and on volcanic zoning.
- **Please give a brief explanation how the zonation helps in the calculation of seismic hazard, ground shaking, etc.**: Done, in the introduction (**l.41**: *Estimation of the probabilistic seismic hazard assessment (PSHA) is based on (\*) identifying earthquake sources, (\*) characterizing their magnitude-frequency distributions, (\*) the distribution of source-to-site distances and (\*) predicting ground motion intensity (Baker et al., 2021). In this study, we focus on the first step by determining earthquake area and fault sources of the Lesser Antilles arc (hereafter "seismotectonic zoning model".*)

**(8) Other comments:**

- **Line 320 – The slab interface is divided into sections, one of them spanning depths of 35-65 km, which according to Figure 7, is the maximum depth of slab interface earthquakes (although this is never being clearly stated anywhere in the manuscript).** In the manuscript, we specified l.341: "*North of Guadeloupe, the interface seismicity is highlighted by moderate M=4.5–6.5 reverse-faulting*

*earthquakes down to 60–65 km depth (Lindner et al., 2023), whereas very few similar events occur to the south.*"

- **On the other hand, the geodetic model defines the high coupling zone to be between 40-80 km (Line 295), where 80 km is close to a temperature of 450°C according to Ezenwaka's model, not 350°. An upper temperature limit of 450° for subduction interface earthquakes was also postulated in Cascadia (e.g., Hyndman and Wang, 1995).** We are simulated here the proposed coupling depths from Philibosian et al. (2022) to try to understand links between geodetic velocities from GNSS and micro-atoll subsidence data as we said l.187: "*Finally, we use the geodetic velocities from van Rijsingen et al. (2021) to calculate geodetic strain rates (**Fig. 4**) and, completed by micro-atoll subsidence data (Philibosian et al., 2022), to test models of megathrust interseismic coupling on 2D cross-sections.*" Moreover, as we said l.359, "*Vertical velocities indicate a general subsidence rate that could, in part, be associated with strong deep (40–70 km) interseismic coupling*" which is not down to 80 km depth. 65 and 70 km depths are quite close. A 5-km difference will not change the experiment at that scale. 65 km well refers to 350°C. Moreover, each of these methods (earthquake locations, interseismic coupling from geodesy or heat-flow and thermal model) are all subject to uncertainties that may explain difference and incoherency between depths. Instrumental earthquake records are an instantaneous screenshot where heat flow measurements, depending from the depth, are expressing long-term thermal processes as heat dissipation by conduction is a relatively slow phenomenon. Moreover, the seismic to aseismic transition may influence these depths. To add clarity, we modify our manuscript as follow: **l.353**: *In order to test the compatibility of Philibosian et al. (2022) and van Rijsingen et al. (2021) results, we performed a first-order simulation of the megathrust interface interseismic coupling depth along a 2D cross-section (**Fig. 8**). We did not consider the 3D slab geometry. We simulated coupling depth proposed by Philibosian et al. (2022) independently to our proposed area sources.*

- **There is very little information about seismic hazard from volcanic activity (Section 3.6), so it can be safely removed. The lack of information can be mentioned in a single sentence in either the Introduction or the Discussion sections.** We understand that information from seismic hazards induced by volcanic activity is sparse, but we consider it useful to keep a note of volcanic activity in the main text, so that this element can be taken into account (at minimum by extracting volcano-seismic events from the main catalog) in future PSHA calculations and studies.

- **It will be helpful to add a generic cross-section of the subduction zone at the beginning of Section 3 showing the locations of the different zones discussed in this section.** We agree this kind of figure may help the reader find their way in space. As we are limited in figure numbers, we add this figure in supplementary material (**S1**) and mention it in the introduction (l.25). We accordingly adjust supplementary material numbering.

  **Annex S1 Subduction zone seismic area sources**: *Cross-section across Dominique island illustrating seismic area sources in relation to the general structural data: upper plate crust (green), mantle wedge (purple), plate interface (light pink) and downgoing plate (dark pink). Moho and slab top (solid black lines) are from Paulatto et al. (2017).*

*The mean moho at 28 km depth (dashed black line) is from Kopp et al. (2011) and Bie et al. (2020). The 65 km depth indicates the plate interface downdip limit. Earthquakes (white and gray dots) are from the ISCU-cat (Bertil, 2024) and CDSA (Massin et al., 2021) respectively.*

[Figure]

- **Outer rise or subduction interface sources proposed from tsunami models (Cordrie et al., 2022, Wei et al. 2024) are not mentioned or considered here. Why?** We are not sure of which events you are referring to. The 1690, 1867, 1950-51, 1969, 1974, 1985 and 2004 events are all mentioned in the text (section 2 or 4.1) or in Table 1, and the 1935 event is mentioned in S3 (these events mentioned in Cordrie et al., 2022)). About the normal faults "neglected" in Caribbean hazard assessments as mentioned by Wei et al., (2024), we cannot consider faults for PSHA without estimated return time period or slip rate (parameters needed for seismic hazard). Moreover, we mentioned them in section 2 and declared in section 4.1.3 that "*Fault geometries and slip rates are required to integrate fault sources in seismic hazard models. Only a few Lesser Antilles faults meet these criterions*". Moreover, the time recurrence for tsunami is not the same as earthquake time recurrence. The author evaluated some scenarii, but we did not see in the conclusion results on slip-rate or time recurrence for a specific giving fault.

**(9) Minor comments:**
- **Table 1 – For Anegada and Muertos – Either cite specific references for these regions not block models or general regional compilations, or delete.** We kept the compilation mentioned by Zimmermam et al. (2022) and refer to the references contained therein.
- **Fig. 4 – What is "AW Death"? Do you mean Inactive accretionary wedge?** In Fig.4 caption, AW corresponds to Accretionary Wedge. The D corresponds to Death. Below Fig.4, in AW description, we precise "The southern end corresponds to a zone of

progressive termination (AWD)", i.e., the progressive disappearance of the accretionary wedge. We modified the sentence as follows: **l.258**: *The southern end corresponds to a zone of progressive termination (AWD), where the accretionary wedge gives way to more compacted material. This boundary is not clearly defined.*